# A Comprehensive Investigation into the Crystallology, Molecule, and Quantum Chemistry Properties of Two New Hydrous Long-Chain Dibasic Ammonium Salts C_n_H_2n+8_N_2_O_6_ (n = 35 and 37)

**DOI:** 10.3390/ijms24065467

**Published:** 2023-03-13

**Authors:** Zengbo Ke, Xinhui Fan, Youying Di, Fengying Chen, Xi Han, Ke Yang, Bing Li

**Affiliations:** 1School of Materials Science and Chemical Engineering, Xi’an Technological University, Xi’an 710021, China; 2College of Chemical Engineering and Modern Materials, Shaanxi Key Laboratory of Comprehensive Utilization of Tailings Resources, Shangluo University, Shangluo 726000, China

**Keywords:** 1,3-propanediamine dihexadecanoate, 1,3-propanediamine diheptadecanoate, dibasic ammonium salts, crystal structure, Hirshfeld surface analysis, DFT

## Abstract

Through the salification reaction of carboxylation, successful attachment of the long-chain alkanoic acid to the two ends of 1,3-propanediamine was realized, which enabled the doubling of the long-chain alkanoic acid carbon chain. Hydrous 1,3-propanediamine dihexadecanoate (abbreviated as 3C16) and 1,3-propanediamine diheptadecanoate (abbreviated as 3C17) were synthesized afterward, and their crystal structures were characterized by the X-ray single crystal diffraction technique. By analyzing their molecular and crystal structure, their composition, spatial structure, and coordination mode were determined. Two water molecules played important roles in stabilizing the framework of both compounds. Hirshfeld surface analysis revealed the intermolecular interactions between the two molecules. The 3D energy framework map presented the intermolecular interactions more intuitively and digitally, in which dispersion energy plays a dominant role. DFT calculations were performed to analyze the frontier molecular orbitals (HOMO–LUMO). The energy difference between the HOMO–LUMO is 0.2858 eV and 0.2855 eV for 3C16 and 3C17, respectively. DOS diagrams further confirmed the distribution of the frontier molecular orbitals of 3C16 and 3C17. The charge distributions in the compounds were visualized using a molecular electrostatic potential (ESP) surface. ESP maps indicated that the electrophilic sites are localized around the oxygen atom. The crystallographic data and parameters of quantum chemical calculation in this paper will provide data and theoretical support for the development and application of such materials.

## 1. Introduction

Long-chain saturated fatty acids play various roles in the new metabolism of animals and plants [1,2,3,4]. Due to their carboxyl group, long-chain saturated fatty acids can undergo an esterification reaction [5,6], acylation reaction [7,8], salt formation reaction [9], oxidation-reduction reactions [10], and decarboxylation reactions [11,12,13]. In recent years, in addition to being widely studied in the field of biochemistry, long-chain saturated fatty acids have also drawn much attention in the field of thermodynamics [14,15], especially in the area of phase change energy storage [16,17,18,19,20,21,22], where they are becoming increasingly popular.

The popularity of long-chain saturated fatty acids is attributed to their desirable properties, such as good cycling stability, no supercooling, and no phase separation [23,24]. This type of phase change material and its composites are mainly applied in solar energy generation [25,26,27,28], industrial waste heat recovery [29,30], automobile exhaust utilization [31,32], and building heat storage [33,34]. Modifying long-chain fatty acids by physical and chemical methods to increase their latent heat of phase transition is a very important research field.

As a nucleophilic reagent, 1, 3-propylene diamine is alkaline and can form hydrogen bonds. It is often used as an intermediate and solvent in organic synthesis [35,36]. In addition, 1,3-propanediamine plays an important role in photosynthesis and the cultivation of biological strains [37,38]. Amines and their derivatives have been widely reported [39,40]. The binary ammonium salt formed with ethylenediamine and lauric acid as ligands was reported in the literature [41]. The results showed that this kind of binary ammonium salt had good thermodynamic properties. However, the synthesis of long-chain binary ammonium salts with 1,3-propanediamine and long-chain fatty acids as ligands has not been reported. The study of binary ammonium salt by quantum chemical calculation [42,43,44,45] has never been reported.

Taking the above into consideration, this paper successfully synthesizes two hydrous long-chain dibasic ammonium salts C_n_H_2n+8_N_2_O_6_ (n = 35 and 37) with hexadecanoic acid, heptadecanoic acid, and 1,3-propanediamine as the raw materials, realizing the doubling of the carbon chain length of long-chain dibasic acid. The molecular structures of two compounds are determined by an X-ray single crystal diffractometer. Their intermolecular interactions and hot spots are revealed by Hirshfeld surface analysis. In addition, their frontier molecular orbitals, chemical reaction parameters, electronic state densities, and molecular surface electrostatic potentials are disclosed by DFT theory and the latest quantum chemistry tools.

## 2. Results and Discussions

### 2.1. Descriptions of Crystal Structure

The crystal size and data obtained by X-ray single crystal diffraction are shown in Table 1. Table 1 shows that the crystal of the compound hydrous 1,3-propanediamine dihexadecanoate (3C16) is triclinic; a space group is P-1 and Z = 2 with unit cell dimensions *a* = 6.6497(8) Å, *b* = 8.4340(9) Å, *c* = 35.221(4) Å, *α* = 90.747(2)°, *β* = 90.748(2)°, and *γ* = 96.969(3)°. The crystal of the compound hydrous 1,3-propanediamine diheptadecanoate (3C17) is triclinic; a space group is P-1 and *Z* = 2 with unit cell dimensions *a* = 6.6942(13) Å, *b* = 8.4899(17) Å, *c* = 37.083(7) Å, *α* = 92.15(3)°, *β* = 93.65(3)°, and *γ* = 97.04(3)°. It can be seen that both crystal systems of the two lactate complexes are triclinic. The two crystal structures have the same crystal system and space group as reported in the literature [41]. Unlike the compound reported in the literature, the number of molecules in a single crystal cell and the lengths of the molecules are different.

Figure 1a,b show the molecular elliptical diagrams of 3C16 and 3C17, respectively, indicating that they are typical amphiphilic molecules. The head hydrophilic polar groups, carboxylate ions and ammonium ions, and the hydrophobic non-polar hydrocarbon chains at the tail are folded. The unit cell diagrams of 3C16 and 3C17 are shown in Figure 2a,b, respectively. It can be seen from the cell diagrams that their spatial arrangement is the same. Hydrophilic groups of both compounds are located inside the cell [41]. This can also be seen from the 2D space stacking diagrams in Figure 3 and the 3D space-filling diagram in Figure 4. Strong hydrogen bonding plays an important role in the orderly arrangement of the two molecules in space. Hydrogen bonds in Figure 2a, where amines act as donors and carboxylates act as receptors, include N1-H1A...O3, N1-H1B...O1, N1-H1C...O4, N2-H2C...O1, N2-H2D...O2, and N2-H2E…O2. Hydrogen bonds where H_2_O acts as donors and carboxylates act as receptors include O5-H5C…O2, O5-H5D…O3, and O6-H6D…O3, and the hydrogen bond where H_2_O acts as donors and acceptors are O6-H6C…O5. In Figure 2b, hydrogen bonds in which amines serve as donors and carboxylates serve as receptors include N1-H1A...O3, N1-H1B...O2, N1-H1C...O4, N2-H2C...O2, and N2-H2D...O1. Hydrogen bonds in which H_2_O acts as donors and carboxylates act as receptors include O5-H5C…O1, O5-H5D…O3, and O6-H6C…O3, and the hydrogen bond in which H_2_O acts as both donor and acceptor are O6-H6D…O5. It can be seen that H_2_O molecules play an important role in stabilizing the framework of the title compounds. The bond lengths and angles of 3C16 and 3C17 are listed in Table 2 and Table 3, respectively, and the hydrogen bond data are listed in Table 4 and Table 5. The 3D space-filling diagrams of 3C16 and 3C17 are shown in Figure 4a,b, respectively. Hydrogen bonding results in the formation of two-dimensional networks of both compounds, which have interpenetrating layers of organic and inorganic components similar to the layered “sandwich” structure found in perovskite [46,47].

### 2.2. Hirshfeld Surface Analysis

Upon inputting the CIF files, CrystalExplorer 17.5 software was used to generate the Hirshfeld surface and 2D fingerprint plot of the title complexes. *d*_e_ and *d*_i_, indicated in the 2D fingerprint plot, refer to the length between the Hirshfeld surface and outermost distance of the closest atom, and the shortest distance between the surface and innermost distance of the closest atom, respectively. *d*_norm_ is a normalized contact distance derived from *d*_e_ and *d*_i_.

Figure 5 and Figure 6 illustrate how an analysis of the 2D fingerprint plots can be employed to detect patterns corresponding to distinct interactions (H...H, H...O, etc.). Figure 5 suggests that compound 3C16 possesses a close H...H interaction (79.4%), as well as H...O (8.8%) and O...H (10.6%) interactions. Similarly, Figure 6 reveals that compound 3C17 is characterized by a close H...H bond (80%), as well as H...O (8.5%) and O...H (10.3%) interactions. The 2D fingerprints of both molecules also point to the fact that the hydrogen bond donor around the carboxyl group is situated beyond the Hirshfeld surface, whereas the hydrogen bond receptor in the vicinity of the carboxyl group is located within the Hirshfeld surface. As a result of the H...O and O...H interactions, both compounds are featured by distinct red-spotted areas on the Hirshfeld surface of the title compounds, consistent with the data presented in Table 4 and Table 5. Intermolecular interactions of the two molecules are mainly impacted by the O-H...O and O-H...O hydrogen bonds.

The mechanical strength of a single crystal is related to the spatial crystal packing. Single crystals with large cavities show a limited capacity for withstanding external forces, whereas those without large cavities exhibit a notable ability to bear considerable forces or stresses [48,49]. We carried out the void analysis on 3C16 and 3C17 crystals, which is based on adding up the atomic electron density by using the Hartree–Fock theory. It is assumed that all the atoms are spherically symmetric while calculating voids. Refer to Appendix A and Figure 7 for detailed void parameters. When the electron density isosurface value is 0.002 au, the void volumes of 3C16 and 3C17 are 214.46 Å3 and 248.87 Å3, respectively. The volume of voids in 3C16 and 3C17 accounts for 10.94% and 11.93% of the total volume, respectively. Since the space occupied by the voids in the two compounds is very small, there is no large cavity in the crystal packing of 3C16 and 3C17. We can speculate that 3C16 and 3C17 have good mechanical properties. 

The ability of a pair of chemical species (X, Y) to form crystal packing interactions is determined by computing the enrichment ratio. The enrichment ratio is calculated by dividing the proportion of the actual contacts by the theoretical proportion of the random contacts [50,51,52]. For a particular crystal, some contacts are more favorable to forming crystal packing interactions than other contacts. The enrichment ratio for a contact provides the tendency of it to form crystal packing interactions. The contacts with an enrichment ratio greater than one have a higher tendency to form crystal packing interactions as compared to other contacts. Appendix A list the enrichment ratios of all possible chemical pairs of 3C16 and C17. From Appendix A, it can be seen that the enrichment ratios of C-H contact, O-H contact, and H-H contact in the 3C16 molecule are 0.83, 1.19, and 0.97. From Appendix A, it can be seen that the enrichment ratios of C-H contact, O-H contact, and H-H contact in the 3C17 molecule are 0.89, 1.18, and 0.97. It can be seen that the O-H contact in the two molecules is beneficial.

### 2.3. Energy Frameworks

The construction of an energy framework provides three-dimensional visualization of the supramolecular assembly within crystal molecules. The energy of molecular interactions is typically represented by four distinct components: electrostatics, polarization, dispersion, and exchange repulsion, expressed as *E*_tot_ = *k*_ele_*E*_ele_ + *k*_pol_*E*_pol_ + *k*_dis_*E*_dis_ + *k*_rep_*E*_rep_ [53]. Using the CrystalExplorer 17.5 software, the energy framework was calculated using the HF method with 3–21G basis set. The energy for molecular interactions was computed using the intermolecular potential method. Three types of intermolecular interaction energies were involved in the energy calculation: electrostatic energy, dispersion energy, and total energy. An energy frame of 2 × 1 × 1 size clusters was generated to calculate the energy. For compounds 3C16 and 3C17, the intermolecular interaction energy frame diagrams along the a, b, and c directions are shown in Figure 8 and Figure 9, respectively. The numerical values of the intermolecular interaction energies involved in the energy calculation are listed in Table 6 and Table 7.

The ratio factors of energy computed using the HF/3–21G basis set were found to be k_ele_ = 1.019, k_pol_ = 0.651, k_dis_ = 0.901, and k_rep_ = 0.811 [54]. Calculations on the data from Table 5 and Table 6 yielded the intermolecular energies for the title compounds; 3C16 had electrostatic, polarization, dispersion, and exchange repulsion energies of 3.5 kJ/mol, −7.4 kJ/mol, −195.5 kJ/mol, and 63.3 kJ/mol, respectively, and 3C17 had electrostatic, polarization, dispersion, and exchange repulsion energies of 4.7 kJ/mol, −7.4 kJ/mol, −196 kJ/mol, and 57 kJ/mol, respectively. The total energies were −126.3 kJ/mol and −130.4 kJ/mol for 3C16 and 3C17, respectively. It can be seen that dispersion energy dominates electrostatic energy in both compounds. The size of the small cylinders in Figure 8 and Figure 9 revealed the strength of intermolecular energy and its correlation to molecular stacking. Note that those weak intermolecular interactions below a certain threshold are omitted to avoid congestion. The absence of cylinders in the energy framework along a particular direction does not necessarily imply the absence of any stabilizing intermolecular interactions.

### 2.4. Quantum Chemical Calculations

#### 2.4.1. Molecular Geometry Optimization

The molecular geometry optimization and frequency calculations of the title compounds were achieved through density functional theory (DFT) [55,56]. DFT is a widely used technique for studying electronic structures in materials science. It is a tool for investigating properties such as geometry optimization, infrared spectra, molecular orbitals, and molecular surface electrostatic potentials.

Density functional theoretical (DFT) computations were performed with Gaussian 09 software [57] using the B3LYP/6–31G* basis set. All quantum chemical calculations presented herein were performed within the context of a periodic system to accurately reflect the crystal environment. Optimized geometries of the title compounds were obtained and the comparison of the experimental structures to the molecular optimized structures is shown in Figure 10, which demonstrates the good consistency between the bond lengths and bond angles for the title compounds. For 3C16, the correlation coefficients are R^2^ = 0.99997 (bond length) and R^2^ = 0.99971 (bond angle), respectively. For 3C17, the correlation coefficients are R^2^ = 0.99998 (bond length) and R^2^ = 0.99984 (bond angle), respectively. Table 2 and Table 3 list the comparisons between the optimized structural parameters, bond lengths, and bond angles for the experimental and calculated results, respectively.

#### 2.4.2. Frontier Molecular Orbitals

Frontier molecular orbitals (FMOs) play a crucial role in predicting the chemical reactivity and stability of molecules [58,59,60]. FMOs refer to the collective term of a molecule’s highest occupied molecular orbital (HOMO) and lowest unoccupied molecular orbital (LUMO), the energy gap (i.e., the gap) between the HOMO and the LUMO reveals the charge transfer of electrons. The gap defines the first excited state, reflecting the dynamical stability and chemical reactivity of the molecule.

Figure 11 shows the energy level diagram of the frontier molecular orbitals and secondary orbitals for 3C16, where the HOMO and LUMO are both distributed in the carboxyl and amido. The HOMO of the secondary orbitals is distributed in the carboxyl group, indicating the nucleophilic region, and the LUMO is located in the amido group, indicating the electrophilic region. Figure 12 shows the energy level diagrams of the frontier molecular orbitals and secondary orbitals for 3C17, whose distribution of the HOMO and the LUMO on the functional groups is the same as that of 3C16. The energy gap of the frontier molecular orbitals for 3C16 is 0.2858 eV and the energy gap of the secondary orbitals is 0.6803 eV. The energy gap of the frontier molecular orbitals for 3C17 is 0.2855 eV and the energy gap of the secondary orbitals is 0.6966 eV.

Through the analysis of frontier molecular orbitals, we can obtain various molecular reactivity descriptors [53] to better understand the chemical properties of the title compounds, where molecular electronegativity (*χ*) and chemical hardness (*η*) of the molecules were calculated using the formula, *χ* = (*I* + *A*)/2, and *η* = (*I* − *A*)/2, where *I* is the ionization energy, which is a measure of the electron giving the ability of the molecules, and *A* is the electron affinity, which is a measure of the electron receiving ability of the molecules. In numerical terms, *I* = −*E*_HOMO_, and *A* = −*E*_LUMO_. Chemical potential (*μ*) is opposite to molecular electronegativity in numerical value, i.e., *μ =* −*χ*. The chemical flexibility (*σ*) and electrophilicity index (*ω*) of the molecules were calculated using the formula, *σ =* 1/2*η* and *ω*
**=**
*χ*^2^/2*η*. The calculated results of the reactivity descriptors of 3C16 and 3C17 are listed in Table 7 and Table 8.

#### 2.4.3. Density of States

The density of states (DOS) is essentially the number of different states of molecular orbitals under a certain energy level [61,62,63], and the corresponding DOS graph is an important analytical tool. TDOS describes the entire system orbits, or with the help of partial density of states (PDOS), contributes to each molecular orbital in the whole system. The overlap population density of states (OPDOS) is useful in examining the interaction between fragments, and its numerical value is positive for covalent bond orbitals and negative for antibonding orbitals.

The DOS analysis of the two molecules was performed using the B3LYP density functional method with 6–31G* basis set, and the plots were drawn by the *Multiwfn* program package. Figure 13a,c depict the DOS of 3C16 and 3C17 drawn by the Hirshfeld method and by different functional groups, respectively. Figure 13a shows that the functional group contributing the most to the HOMO (−2.9334 eV) of 3C16 is carboxyl, followed by amine. Figure 13c reveals that the functional group contributing the most to the HOMO (−2.9943 eV) of 3C17 is carboxyl, followed by amine, which is consistent with the results of Section 2.4.2. The projection of the density of states (PDOS) analysis, as depicted in Figure 13a,c, elucidates the electronic contribution of various functional groups to the frontier molecular orbitals. Notably, the amine groups, upon protonation, exhibit a significant shift in their electronic density, which is reflected in the negative LUMO energy values. The protonation of the amine nitrogen leads to a withdrawal of electron density from the molecular orbitals, resulting in an increase in the LUMO energy. This effect is more pronounced in the crystalline state due to the periodic potential and the interactions with neighboring molecules, which is consistent with the observed negative LUMO energy values. Figure 13b,d shows the DOS of 3C16 and 3C17 plotted according to different angular momenta. Figure 13b,d indicate that the orbitals contributing the most to the HOMO of 3C16 and 3C17 are *p* orbitals, followed by *s* orbitals. The analysis of OPDOS results shows that the antibonding orbitals of the two compounds appear in approximately the same position.

#### 2.4.4. Molecular Electrostatic Potential

The molecular electrostatic potential (ESP) is a crucial concept in wavefunction analysis [64,65,66,67], playing a key role in discussions of electrostatic interactions. ESP analysis helps to identify reactive sites in molecules, which are determined by their electrostatic potential values computed for uniformly distributed regions on a van der Waals surface. The molecular electrostatic potential V(*r*) at each point *r* in the surrounding space is generated by the electron and atomic nucleus of the molecule, V(*r*) = *Z*_A_/(*R*_A_ − *r*) − ∫*ρ*(*r*’)d *r*’/|(*r* − *r*’)|, where *Z*_A_ is the charge at radius *R*_A_ on the atomic nucleus *A*, and *ρ*(*r*) is the electron density of the molecule. The ESP map was drawn with a combination of Gaussian 09, Multiwfn package, and VMD software [68], based on the B3LYP density functional method and 6–31G* basis set.

Figure 14a,b show the molecular surface electrostatic potentials of 3C16 and 3C17, respectively. The red (positive) coloration area on the ESP map indicates the hydro-positive sites, while the blue (negative) coloration area indicates the electro-positive sites. The negative electro-positive sites of 3C16 mainly focus on carboxylic, with a minimum electrostatic potential of −113.50 kcal/mol. The positive electro-positive sites of 3C16 are dispersed around amine, with a maximum electrostatic potential of 110.44 kcal/mol. The distribution of positive and negative electro-positive sites of 3C17 is the same as 3C16, with a minimum electrostatic potential of −112.54 kcal/mol and a maximum electrostatic potential of 110.82 kcal/mol. The detailed data of the electrostatic potential distribution of 3C16 and 3C17 are listed in Appendix A. The large difference in electrostatic potential between the two molecules can be used to predict that their active sites can interact strongly with adjacent molecules. Figure 14c,d show the quantitative distribution of the molecular surface electrostatic potential of 3C16 and 3C17. It can be seen from the chart that the molecular surface electrostatic potential of these two molecules is mainly focused between −20~20 kcal/mol. Most of them are near 0 kcal/mol, which is powerful evidence of weak intermolecular and intramolecular interactions.

## 3. Materials Methods

### 3.1. Sample Synthesis and Instruments

All reagents and solvents required for the synthesis were purchased from commercial suppliers in China. Specifically, 1,3-propanediamine, hexadecanoic acid, heptadecanoic acid, and ethanol were all of analytical purity, with specifications above 95%. In the experiment, the molar ratio of the two ligand acids to 1,3-propanediamine was 2.5:1. The excess of acid was chosen to ensure that the carboxyl groups fully engage with the two amine groups. The key step in synthesizing 3C16 involved weighing out 0.0038 mol of solid hexadecanoic acid and dissolving it in 100 mL of ethanol solution to form a precursor clear solution for 3C16. To this precursor, 0.0015 mol of 1,3-propanediamine liquid was added dropwise, with the strict condition that no precipitate was allowed to form during the slow addition process. The well-prepared clear solution was then heated on a rotary evaporator with a rotation speed of 1500 r/min and a temperature of 60 °C. Heating was stopped when the solution volume was reduced to 60 mL, and the power was turned off. The key steps for synthesizing 3C17 were similar: 0.0038 mol of solid heptadecanoic acid was weighed out and dissolved in 100 mL of ethanol solution to form a precursor clear solution for 3C17. Again, 0.0015 mol of 1,3-propanediamine liquid was added dropwise, with the same condition regarding the absence of precipitate. Under the same conditions, the well-prepared clear solution was heated on the rotary evaporator, and heating was ceased when the solution volume was reduced to 80 mL, followed by power shutdown.

The post-reaction solution needed to be slowly cooled at room temperature, awaiting the crystallization of the sample. The crystallized samples were then subjected to recrystallization treatment. After drying in a vacuum desiccator for six days, the samples were placed in weighing bottles for further use. The actual mass fraction purity of samples 3C16 and 3C17, as determined by high-performance liquid chromatography (HPLC), was above 98%. Calculations revealed that the yields of both compounds were over 40%. It should be noted that the reaction yield was not the most critical factor, as we had to perform repeated recrystallization on the primary products, at least three times or more. The scheme of the synthesized compounds is shown in Figure 15. Agilent GC 6890N was used for gas chromatography, Vario EL III was used for element analysis, and XD-2700 was used for XRD.

### 3.2. Basic Experimental Data

Hydrous 1,3-propanediamine dihexadecanoate (abbreviated as 3C16) and 1,3-propanediamine diheptadecanoate (abbreviated as 3C17) were successfully synthesized. Translucent colorless solid, yield: 3C16: 82% (214 mg); 3C17: 75% (192 mg). XRD (Cu-Kα1 radiation, λ = 0.15406 nm): 3C16: (0 0 3), (1 0 4), (0 1 6), (1 0 
$\bar{7}$
), (1 
$\bar{2}$
 0), (1 
$\bar{2}$

$\bar{2}$
); 3C17: (0 0 3), (1 0 4), (1 0 5), (1 0 6), (1 
$\bar{2}$
 1), (1 
$\bar{2}$

$\bar{2}$
). The XRD diagrams are shown in Figure 16. Elemental analysis calcd (%) for 3C16 (622.99): C, 67.19; N, 4.42; H, 12.75; O, 15.64; found: C, 67.48; N, 4.50; H, 12.62; O, 15.40. Elemental analysis calcd (%) for 3C17 (651.04): C, 68.02; N, 4.23; H, 12.84; O, 14.91; found: C, 68.26; N, 4.30; H, 12.70; O, 14.74. The oxygen atoms’ content was measured by indirect method.

### 3.3. X-ray Crystallography

The crystals were glued to the fine glass fibers and then mounted on the Bruker Smart-1000 CCD diffractometer with Mo-Kα radiation, λ = 0.71073 Å. The intensity data were collected in the φ–ω scan mode at T = 273 K. The size of 3C16 is 0.44 × 0.18 × 0.07 mm^3^. The size of 3C17 is 0.12 × 0.11 × 0.1 mm^3^. The structures of title compounds were solved by the direct method and the differential Fourier synthesis, and all non-hydrogen atoms were refined anisotropically on F2 by the full-matrix least-squares method. All calculations were performed with the program package SHELXTL [69]. The program used in the building structure was Diamond 3.2 software (Copyright© 1997–2009 by CRYSTAL IMPACT Dr. K. Brandenburg & Dr. H. Putz GbR). We only needed to import the refined CIF into the software for processing. The relevant atomic theories were hydrogenated and refined. The hydrogen atoms were added theoretically, riding on the concerned atoms, and not refined.

The crystal data and structure refinement for the title compounds are summarized in Table 1. We applied two compounds of 3C16 and 3C17 to the Cambridge crystal data center (CCDC) with numbers 2238301 and 2238306.

### 3.4. CrystalExplorer

In Section 2.3, the CIF format files of title compounds were obtained by the program package SHELXTL. By inputting the CIF files into relevant quantitative calculation software, the weak interaction between complex molecules can be obtained. The graphics software selected for quantum chemical calculation in this experiment was CrystalExplorer 17.5 [70]. CrystalExplorer 17.5 provides a new way of visualizing molecular crystals using the full suite of Hirshfeld surface tools [71]. Hirshfeld surface is the isosurface with a weight coefficient *w*(r) equal to 0.5. The average charge density of molecules inside the isosurface should exceed the average charge density of all surrounding molecules (*w*(r) ≤ 0.5 within the isosurface, *w*(r) ≥ 0.5 outside the isosurface). This ratio is also approximately the ratio of the charge density of real molecules to that of real crystals. Hirshfeld surface [71] is a new definition of molecular surface. Hirshfeld surface analysis can achieve real and continuous 3D visualization, and 2D fingerprint is the two-dimensional representation of Hirshfeld surface analysis.

### 3.5. Multiwfn

*Multiwfn*, fully known as multifunctional wave function analyzer, is a very powerful wave function analysis program written by Chinese scientist Lu Tian [72], which can realize almost all the most important wave function analysis methods in the field of quantum chemistry. *Multiwfn* has the advantages of being easy to learn and use, efficient, flexible, open source, and free. This program has users all over the world and has been used by more than 1000 academic papers or books.

## 4. Conclusions

3C16 and 3C17 belong to the triclinic system with a space group P-1. It was discovered that H2O plays a vital role in securing the molecular framework of the two molecules. Hirshfeld surface analysis verified the presence of N-H...O intermolecular interaction with the amine donor and O-H...O intermolecular interaction with the H2O donor in both of the molecules. The 2D fingerprint indicated that the major contributions come from H...H (3C16 79.4%, and 3C17 80%) bonds. The void analysis showed that the mechanical properties of the two molecules are strong. The enrichment analysis indicated that these two kinds of intramolecular O-H contacts are powerful. A 3D energy framework construction revealed that dispersion energy was predominant in the two molecules. DFT calculations indicated that the experimental structural parameters are consistent with their theoretical counterparts. FMO analysis was used to determine the reactivity descriptors of the two molecules, and the charge distributions on the ESP diagrams demonstrate the chemical reaction sites of the two molecules.

## Figures and Tables

**Figure 1 ijms-24-05467-f001:**
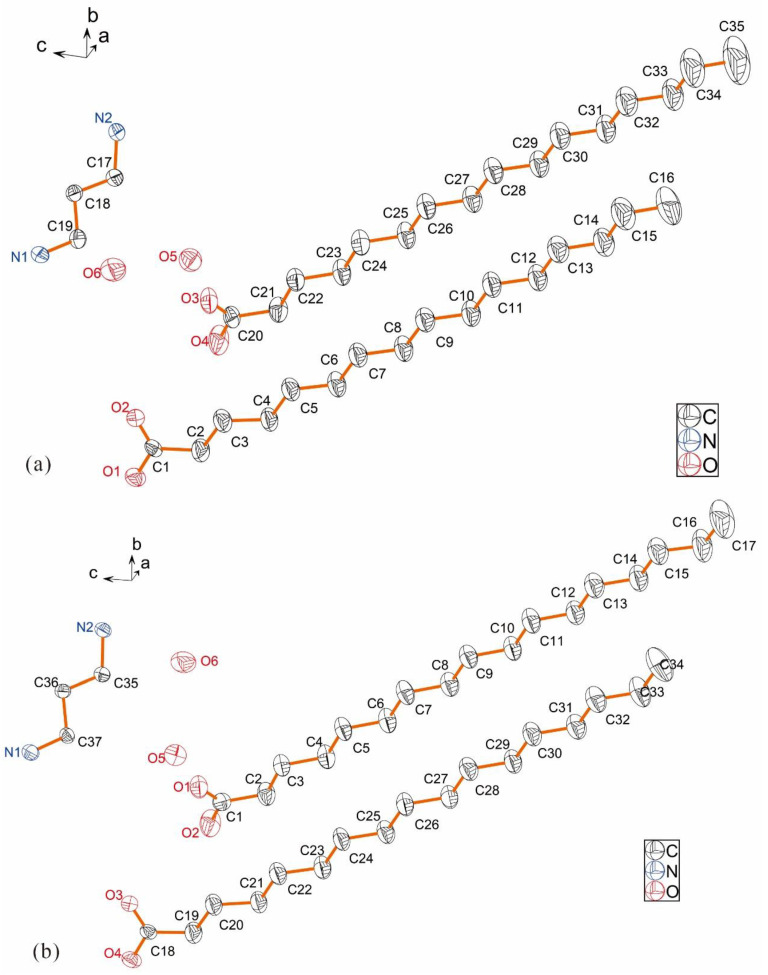
Ellipsoid diagrams of molecular structure of the two molecules: where (**a**) represents 3C16 and (**b**) represents 3C17.

**Figure 2 ijms-24-05467-f002:**
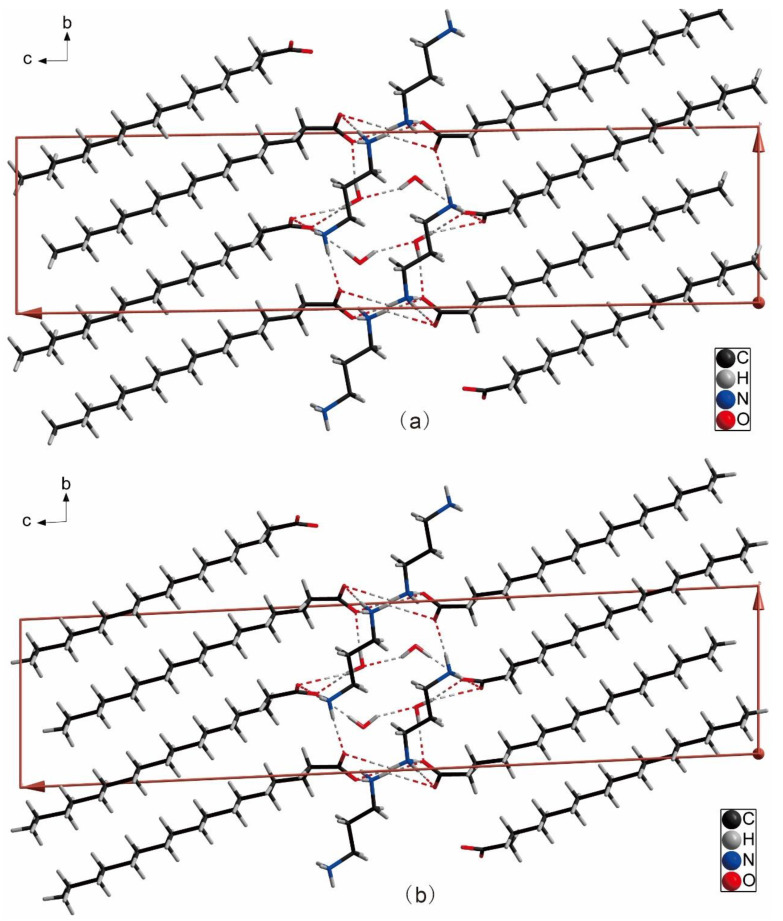
Unit cell diagrams of the two molecules: where (**a**) represents 3C16 and (**b**) represents 3C17.

**Figure 3 ijms-24-05467-f003:**
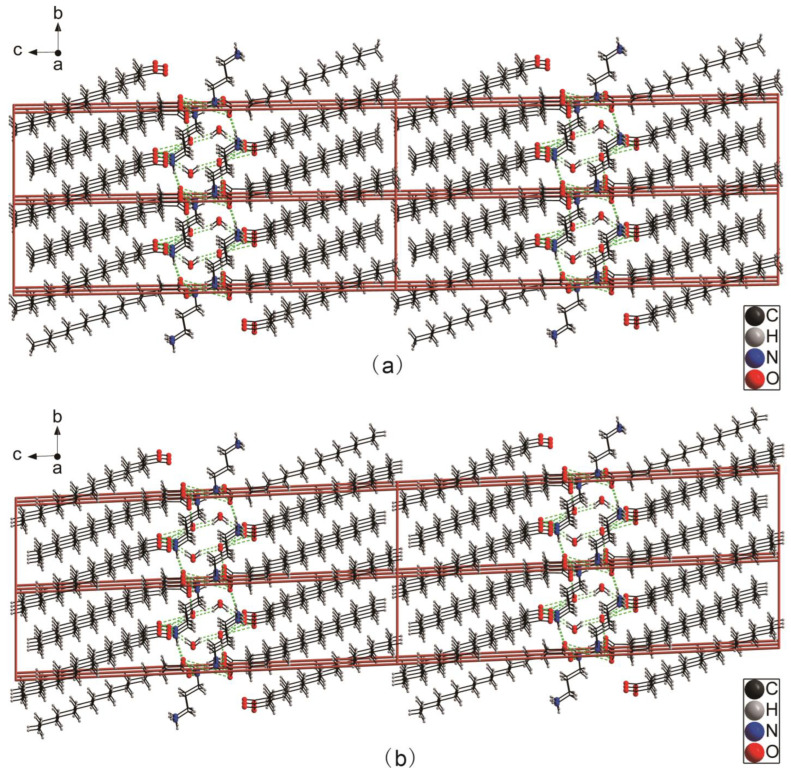
Three-dimensional stacking diagrams (2 × 2 × 2) of the two molecules: where (**a**) represents 3C16 and (**b**) represents 3C17.

**Figure 4 ijms-24-05467-f004:**
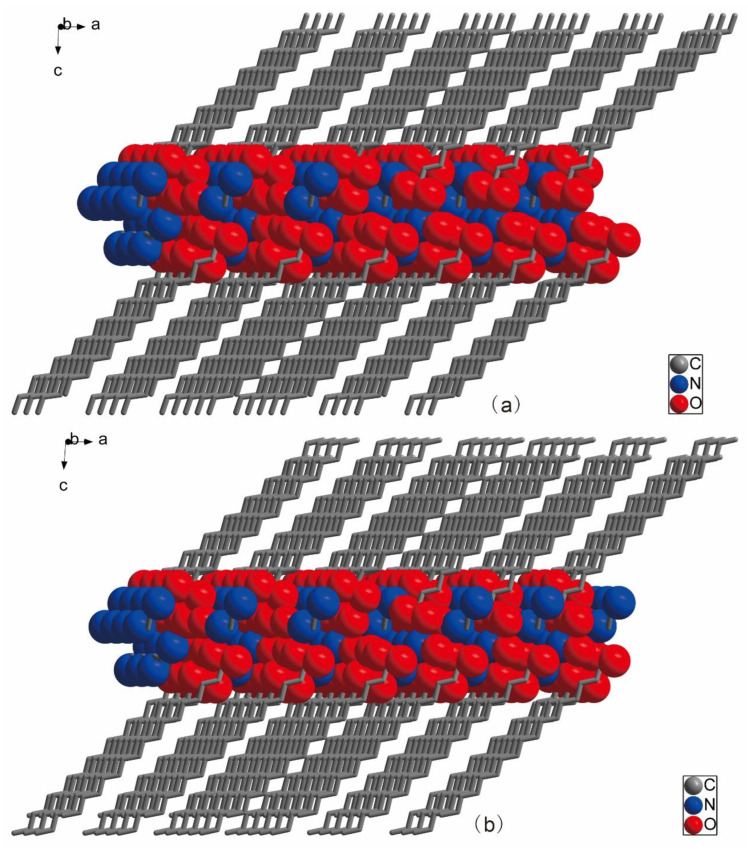
Three-dimensional space-filling diagrams of the two molecules: where (**a**) represents 3C16 and (**b**) represents 3C17.

**Figure 5 ijms-24-05467-f005:**
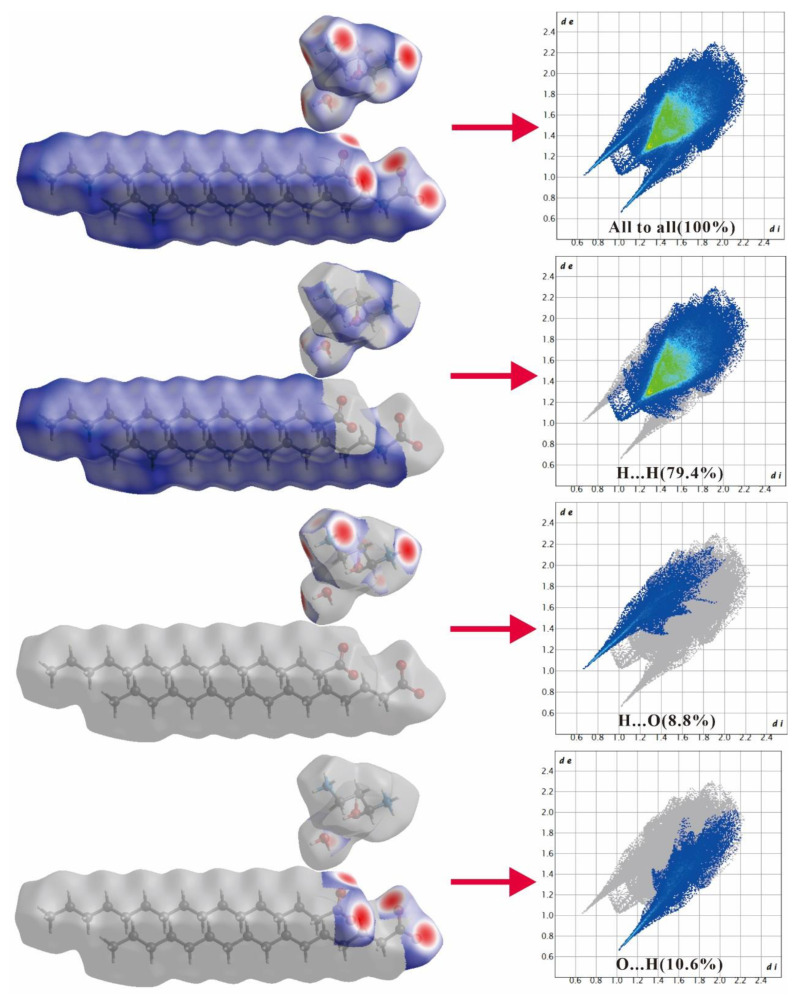
Hirshfeld surface and 2D fingerprint plots for H in molecule 3C16, broken down into contributions from specific pairs of atom types.

**Figure 6 ijms-24-05467-f006:**
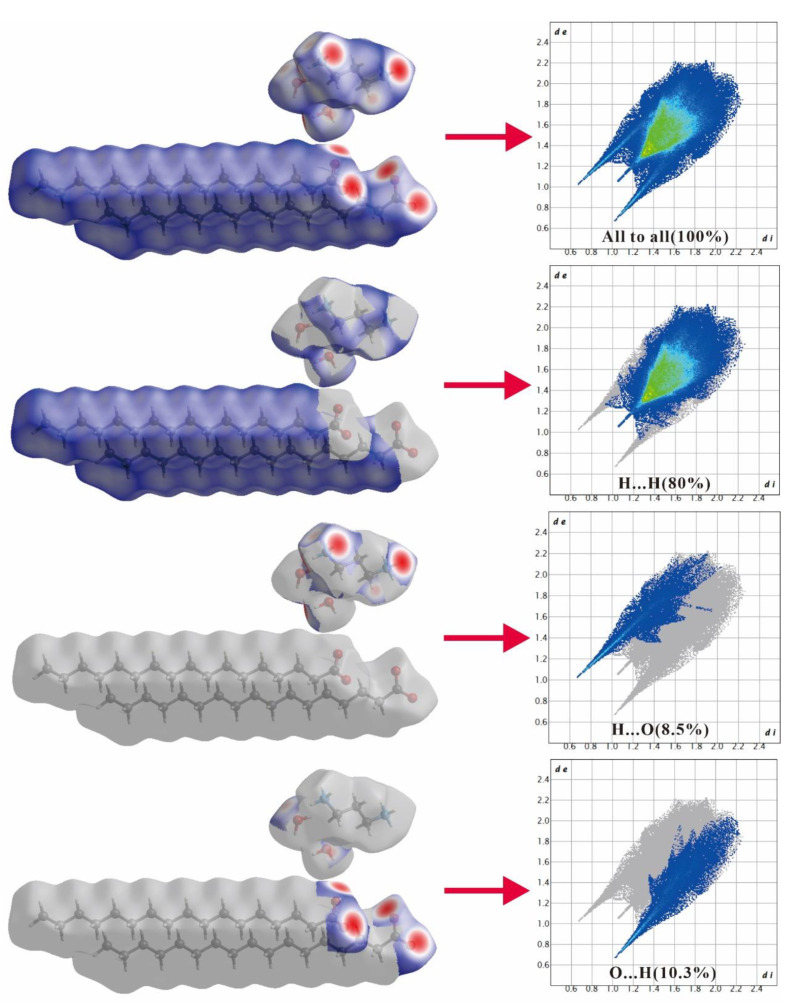
Hirshfeld surface and 2D fingerprint plots for H in molecule 3C17, broken down into contributions from specific pairs of atom types.

**Figure 7 ijms-24-05467-f007:**
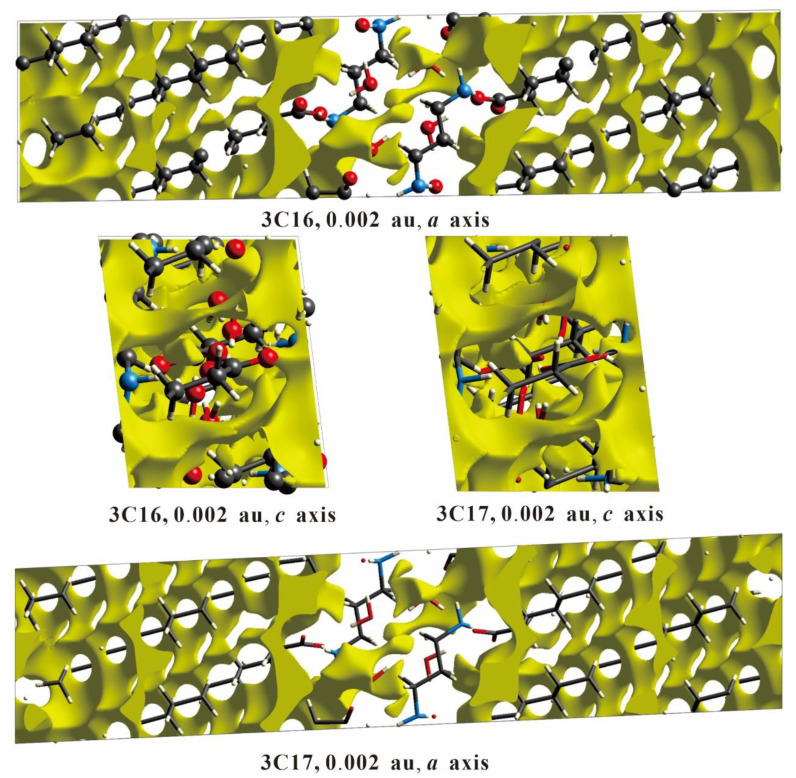
Void surfaces (0.002 au) for 3C16 and 3C17. The view is along the b axis and c axis in all cases, and all diagrams are on the same scale.

**Figure 8 ijms-24-05467-f008:**
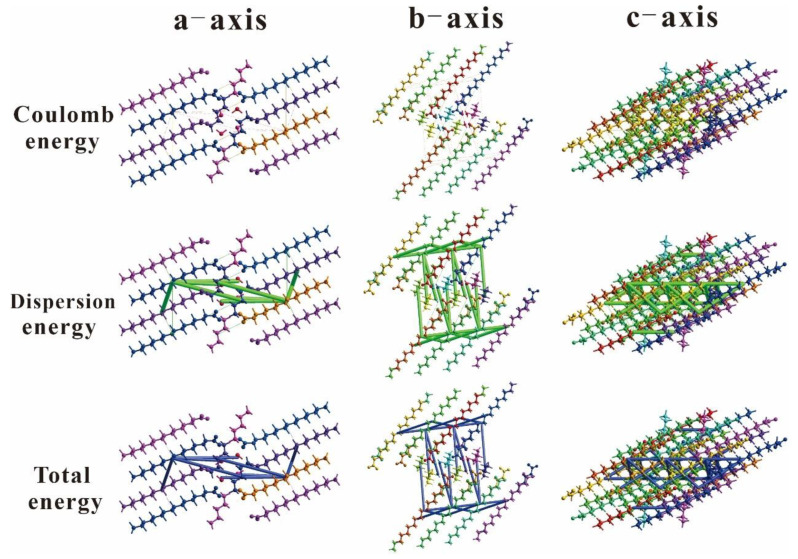
The diagrams of the interaction energies: Coulomb energy, dispersion energy, and total energy of 3C16 molecule along the a, b, and c axes.

**Figure 9 ijms-24-05467-f009:**
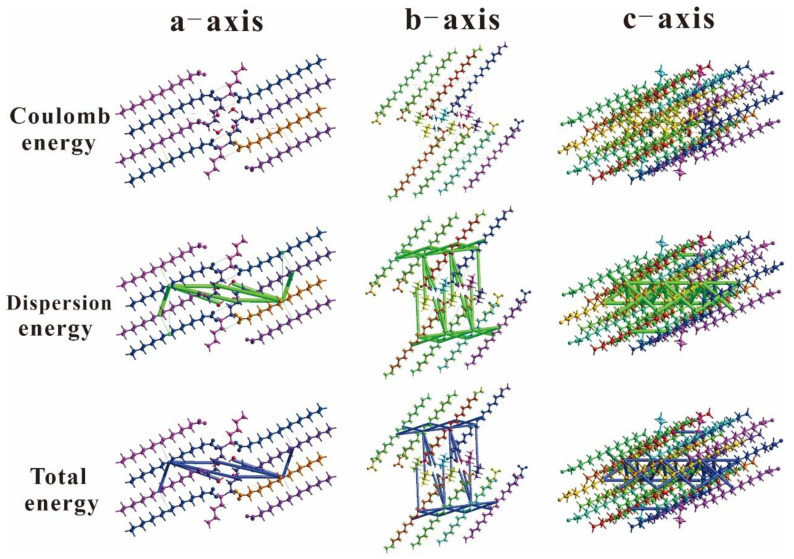
The diagrams of the interaction energies: Coulomb energy, dispersion energy, and total energy of 3C17 molecule along the a, b, and c axes.

**Figure 10 ijms-24-05467-f010:**
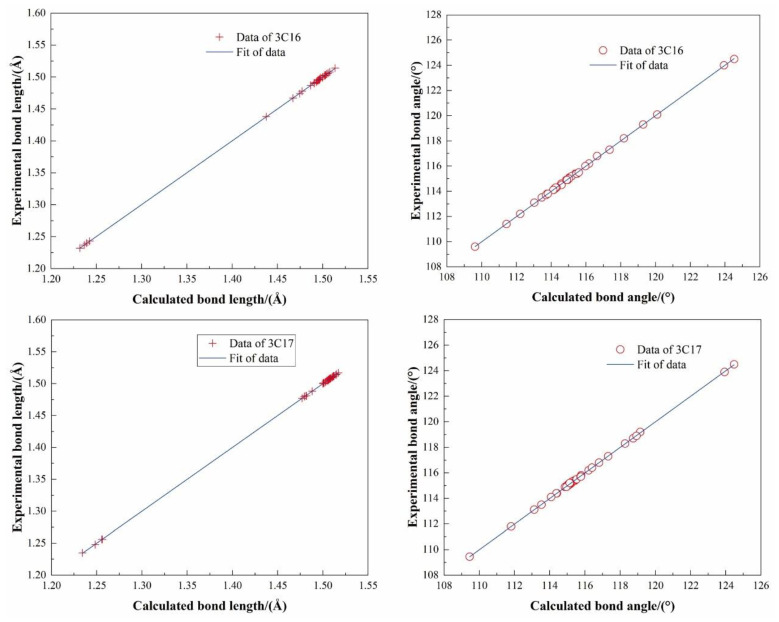
The correlation diagrams between the experimental bond length/angle of the title compounds by X-ray single crystal diffraction and the calculated bond length/angle by DFT.

**Figure 11 ijms-24-05467-f011:**
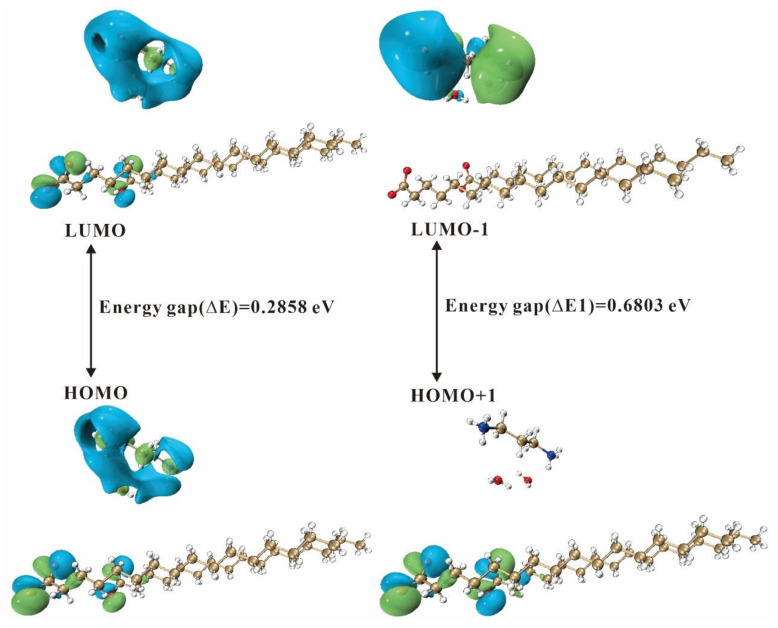
Energy level diagram of the 3C16 molecule.

**Figure 12 ijms-24-05467-f012:**
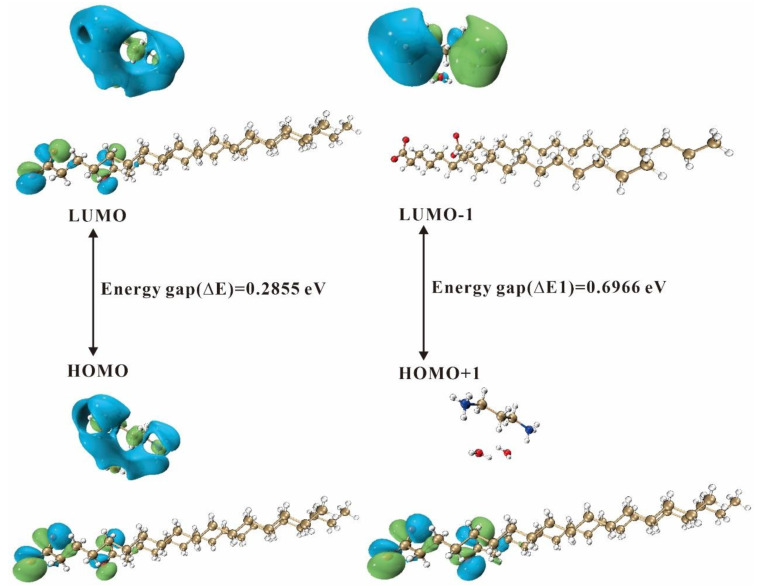
Energy level diagram of the 3C17 molecule.

**Figure 13 ijms-24-05467-f013:**
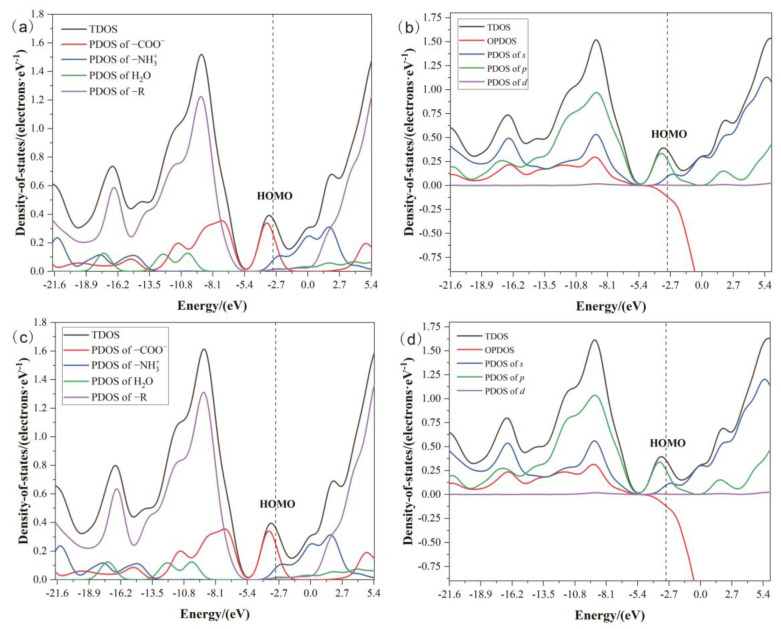
The density of states distribution of the title compounds: (**a**) calculated total DOS and PDOS of different functional groups of 3C16; (**b**) calculated total DOS and PDOS of different angular momentum of 3C16; (**c**) calculated total DOS and PDOS of different functional groups of 3C17; (**d**) calculated total DOS and PDOS of different angular momentum of 3C17.

**Figure 14 ijms-24-05467-f014:**
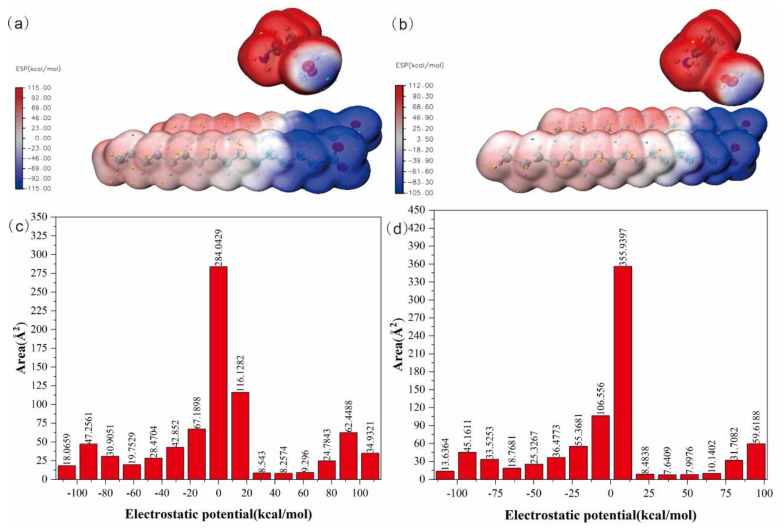
Molecular surface electrostatic potential diagrams and molecular surface electrostatic potential quantitative distribution diagrams of the title compounds: (**a**) molecular surface electrostatic potential diagram of 3C16; (**b**) molecular surface electrostatic potential diagram of 3C17; (**c**) molecular surface electrostatic potential quantitative distribution diagram of 3C16; (**d**) molecular surface electrostatic potential quantitative distribution diagram of 3C17.

**Figure 15 ijms-24-05467-f015:**
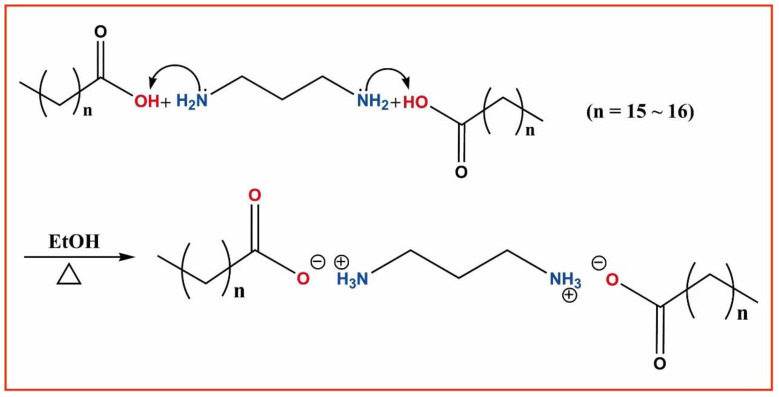
Schematic diagram of the synthesis of the two molecules: where (**top**) represents 3C16 and (**bottom**) represents 3C17.

**Figure 16 ijms-24-05467-f016:**
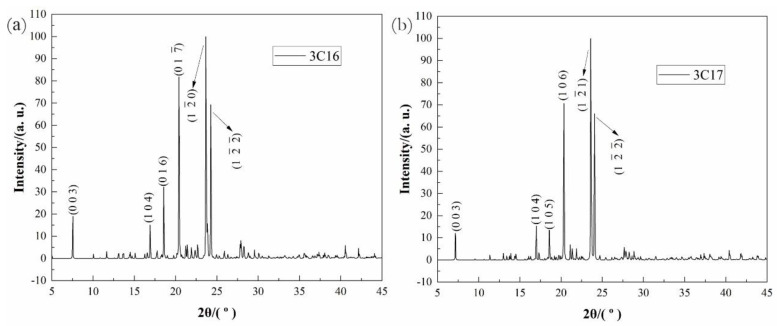
X-ray diffraction patterns of samples the two molecules: where (**a**) represents 3C16 and (**b**) represents 3C17.

**Table 1 ijms-24-05467-t001:** Crystal data and structure refinement for 3C16 and 3C17.

Empirical Formula	C_35_H_78_N_2_O_6_	C_37_H_82_N_2_O_6_
Formula weight	622.99	651.04
Temperature/K	273.15	273(2)
Crystal system	triclinic	triclinic
Space group	P-1	P-1
a/Å	6.6497(8)	6.6942(13)
b/Å	8.4340(9)	8.4899(17)
c/Å	35.221(4)	37.083(7)
α/°	90.747(2)	92.15(3)
β/°	90.748(2)	93.65(3)
γ/°	96.969(3)	97.04(3)
Volume/Å^3^	1960.4(4)	2085.2(7)
Z	2	2
*ρ*_calc_ g/cm^3^	1.055	1.037
μ/mm^−1^	0.07	0.068
F(000)	700	732
Crystal size/mm^3^	0.44 × 0.18 × 0.07	0.12 × 0.11 × 0.1
Radiation	MoKα (λ = 0.71073)	MoKα (λ = 0.71073)
2θ range for data collection/°	4.626 to 48.998	5.984 to 55.108
Index ranges	−7 ≤ h ≤ 7, −9 ≤ k ≤ 9, −35 ≤ l ≤ 41	−8 ≤ h ≤ 8, −11 ≤ k ≤ 11, −48 ≤ l ≤ 48
Reflections collected	19,122	44,383
Independent reflections	6446 (Rint = 0.0382, Rsigma = 0.0711)	9598 (Rint = 0.1409, Rsigma = 0.1786)
Data/restraints/parameters	6446/0/392	9598/0/410
Goodness-of-fit on F^2^	1.12	0.991
Final R indexes [I ≤ 2σ (I)]	R1 = 0.1128, wR2 = 0.2502	R1 = 0.0829, wR2 = 0.1593
Final R indexes (all data)	R1 = 0.1765, wR2 = 0.2793	R1 = 0.2541, wR2 = 0.2113
∆ρ_max_/∆ρ_min_, e/Å^3^	0.21/−0.30	0.25/−0.19

**Table 2 ijms-24-05467-t002:** Bond lengths (Å) and angles (°) for 3C16.

Atoms	Bond Lengths (Å)		Atoms	Bond Angles (°)	
	SCXRD	DFT		SCXRD	DFT
C1–C2	1.505(8)	1.5047	O1–C1–C2	116.2(5)	116.18
C1–O1	1.237(6)	1.2369	O1–C1–O2	124.5(5)	124.53
C1–O2	1.243(7)	1.2428	O2–C1–C2	119.3(5)	119.29
C2–C3	1.478(9)	1.4772	C3–C2–C1	117.3(6)	117.36
C3–C4	1.500(8)	1.5001	C2–C3–C4	115.4(6)	115.39
C4–C5	1.503(8)	1.503	C3–C4–C5	113.8(6)	113.77
C5–C6	1.494(8)	1.4941	C6–C5–C4	115.1(6)	115.06
C6–C7	1.496(9)	1.4965	C5–C6–C7	114.6(6)	114.58
C7–C8	1.495(8)	1.495	C8–C7–C6	114.9(6)	114.95
C8–C9	1.494(9)	1.4941	C9–C8–C7	115.2(6)	115.19
C9–C10	1.502(8)	1.5023	C8–C9–C10	115.4(6)	115.45
C10–C11	1.494(9)	1.4939	C11–C10–C9	114.2(6)	114.22
C11–C12	1.502(9)	1.5021	C10–C11–C12	115.0(6)	115.02
C12–C13	1.491(9)	1.4906	C13–C12–C11	114.5(6)	114.60
C13–C14	1.500(9)	1.4999	C12–C13–C14	115.4(7)	115.48
C14–C15	1.487(11)	1.4867	C15–C14–C13	113.8(7)	113.77
C15–C16	1.492(12)	1.4930	C14–C15–C16	114.9(9)	114.87
C17–C18	1.496(7)	1.4961	N2–C17–C18	113.5(4)	113.46
C17–N2	1.467(7)	1.4672	C17–C18–C19	109.6(4)	109.62
C18–C19	1.499(7)	1.4999	N1–C19–C18	111.4(4)	111.43
C19–N1	1.474(6)	1.4747	O3–C20–C21	120.1(5)	120.10
C20–C21	1.506(8)	1.5061	O4–C20–C21	116.0(6)	115.97
C20–O3	1.240(7)	1.2394	O4–C20–O3	124.0(6)	123.94
C20–O4	1.232(7)	1.2319	C22–C21–C20	118.2(5)	118.18
C21–C22	1.495(8)	1.4952	C21–C22–C23	112.2(5)	112.22
C22–C23	1.514(8)	1.5136	C24–C23–C22	114.1(5)	114.14
C23–C24	1.503(9)	1.503	C23–C24–C25	113.1(5)	113.03
C24–C25	1.503(8)	1.5028	C26–C25–C24	114.9(6)	114.91
C25–C26	1.491(9)	1.4908	C25–C26–C27	114.3(6)	114.31
C26–C27	1.508(8)	1.5079	C28–C27–C26	115.4(6)	115.45
C27–C28	1.497(9)	1.4968	C27–C28–C29	114.2(6)	114.18
C28–C29	1.503(9)	1.5025	C28–C29–C30	113.7(6)	113.70
C29–C30	1.503(9)	1.503	C31–C30–C29	114.3(6)	114.26
C30–C31	1.502(9)	1.5023	C32–C31–C30	114.1(7)	114.12
C31–C32	1.490(10)	1.4903	C31–C32–C33	115.5(7)	115.59
C32–C33	1.499(10)	1.4975	C34–C33–C32	113.8(8)	113.81
C33–C34	1.496(12)	1.4963	C35–C34–C33	116.8(10)	116.64
C34–C35	1.438(14)	1.4375			

**Table 3 ijms-24-05467-t003:** Bond lengths (Å) and angles (°) for 3C17.

Atoms	Bond Lengths (Å)		Atoms	Bond Angles (°)	
	SCXRD	DFT		SCXRD	DFT
C1–C2	1.510(3)	1.5105	O1–C1–C2	118.7(2)	118.74
C1–O1	1.256(3)	1.2563	O2–C1–C2	117.3(2)	117.32
C1–O2	1.248(3)	1.2484	O2–C1–O1	123.9(2)	123.93
C2–C3	1.480(3)	1.4794	C3–C2–C1	118.3(2)	118.28
C3–C4	1.508(3)	1.5080	C2–C3–C4	116.2(2)	116.21
C4–C5	1.501(3)	1.5013	C5–C4–C3	114.4(2)	114.41
C5–C6	1.506(3)	1.5058	C4–C5–C6	115.4(2)	115.44
C6–C7	1.503(3)	1.5031	C7–C6–C5	115.1(2)	115.13
C7–C8	1.512(3)	1.5121	C6–C7–C8	115.1(2)	115.15
C8–C9	1.501(3)	1.5003	C9–C8–C7	115.2(2)	115.22
C9–C10	1.505(3)	1.5046	C8–C9–C10	115.4(2)	115.39
C10–C11	1.507(3)	1.5067	C9–C10–C11	115.3(2)	115.26
C11–C12	1.514(3)	1.5144	C10–C11–C12	115.2(2)	115.24
C12–C13	1.508(3)	1.5075	C13–C12–C11	115.0(2)	115.02
C13–C14	1.510(3)	1.5105	C12–C13–C14	114.4(2)	114.38
C14–C15	1.503(4)	1.5028	C15–C14–C13	115.1(2)	115.12
C15–C16	1.501(4)	1.5006	C16–C15–C14	114.9(3)	114.87
C16–C17	1.511(4)	1.5111	C15–C16–C17	115.1(3)	115.06
C18–C19	1.508(3)	1.5083	N2–C18–C19	113.12(18)	113.1170
C18–N2	1.481(3)	1.4818	C18–C19–C20	109.45(18)	109.4540
C19–C20	1.517(3)	1.5171	N1–C20–C19	111.81(17)	111.8120
C20–N1	1.477(3)	1.4770	O3–C21–C22	119.2(2)	119.14
C21–C22	1.511(3)	1.5111	O4–C21–C22	116.4(2)	116.40
C21–O3	1.256(3)	1.2557	O4–C21–O3	124.5(2)	124.47
C21–O4	1.235(3)	1.2342	C23–C22–C21	118.9(2)	118.93
C22–C23	1.505(3)	1.5056	C22–C23–C24	113.5(2)	113.53
C23–C24	1.515(3)	1.5149	C25–C24–C23	115.5(2)	115.53
C24–C25	1.500(3)	1.5001	C24–C25–C26	114.1(2)	114.08
C25–C26	1.509(3)	1.5091	C27–C26–C25	115.8(2)	115.79
C26–C27	1.505(3)	1.5044	C26–C27–C28	115.1(2)	115.11
C27–C28	1.506(3)	1.5064	C27–C28–C29	115.7(2)	115.75
C28–C29	1.507(3)	1.507	C28–C29–C30	115.2(2)	115.18
C29–C30	1.508(3)	1.5083	C31–C30–C29	115.2(2)	115.15
C30–C31	1.503(3)	1.5033	C30–C31–C32	115.0(2)	114.99
C31–C32	1.512(3)	1.5115	C33–C32–C31	115.0(2)	115.04
C32–C33	1.507(3)	1.507	C34–C33–C32	114.9(2)	114.95
C33–C34	1.505(3)	1.5049	C33–C34–C35	115.7(3)	115.77
C34–C35	1.506(4)	1.5059	C36–C35–C34	115.2(3)	115.14
C35–C36	1.488(4)	1.4881	C37–C36–C35	116.8(3)	116.80
C36–C37	1.481(4)	1.4814			

**Table 4 ijms-24-05467-t004:** Hydrogen Bonds for 3C16.

D-H…A	d(D-H)/Å	d(H-A)/Å	d(D-A)/Å	D-H-A/°
N1-H1A…O3 ^i^	0.89	1.89	2.756(7)	162.8
N1-H1B…O1 ^ii^	0.89	1.91	2.797(6)	175.4
N1-H1C…O4 ^iii^	0.89	1.8	2.687(7)	172.2
N2-H2C…O1 ^iii^	0.89	1.93	2.811(6)	172.8
N2-H2D…O2 ^iv^	0.89	1.91	2.800(6)	174.2
N2-H2E…O2 ^i^	0.89	2.13	2.976(6)	157.8
O5-H5C…O2 ^v^	0.85	1.91	2.745(6)	168.1
O5-H5D…O3 ^v^	0.85	1.99	2.798(5)	158.5
O6-H6C…O5	0.85	2.06	2.863	157.2
O6-H6D…O3 ^i^	0.85	2.07	2.867(6)	156.4

^i^: −*x*, 1 − *y*, 1 − *z*; ^ii^: −1 − *x*, −*y*, 1 − *z*; ^iii^: −1 − *x*, 1 − *y*, 1 − *z*; ^iv^: 1 + *x*, 1 + *y*, +*z*; ^v^: 1 + *x*, +*y*, +*z*.

**Table 5 ijms-24-05467-t005:** Hydrogen bonds for 3C17.

D-H…A	d(D-H)/Å	d(H-A)/Å	d(D-A)/Å	D-H-A/°
N1-H1A…O3 ^i^	0.89	1.91	2.773(3)	161.5
N1-H1B…O2 ^ii^	0.89	1.92	2.806(2)	175.9
N1-H1C…O4 ^iii^	0.89	1.82	2.705(3)	174.6
N2-H2C…O2 ^iii^	0.89	1.94	2.824(3)	170.3
N2-H2D…O1 ^iv^	0.89	1.93	2.813(2)	169.2
O5-H5C…O1 ^v^	0.85	1.93	2.773(2)	172.4
O5-H5D…O3 ^v^	0.85	1.99	2.823(2)	166.4
O6-H6C…O3 ^v^	0.85	2.05	2.891(3)	168.5
O6-H6D…O5 ^vi^	0.85	2.09	2.872(3)	152.6

^i^: −*x*, 1 − *y*, 1 − *z*; ^ii^: −1 − *x*, −*y*, 1 − *z*; ^iii^: −1 − *x*, 1 − *y*, 1 − *z*; ^iv^: 1 + *x*, 1 + *y*, +*z*; ^v^: 1 + *x*, +*y*, +*z*; ^vi^: 1 − *x*, 1 − *y*, 1 − *z*.

**Table 6 ijms-24-05467-t006:** Interaction energies of the 3C16 molecular pairs involved in energy calculation in kJ/mol. R is the distance between molecular centroids in Å, and N is the number of molecular pairs involved.

N	Symop	R	E_ele	E_pol	E_dis	E_rep	E_tot
1	x, y, z	8.43	1.5	−0.1	−1.6	0	0
1	−x, −y, −z	18.75	0	−0.3	0	0	−0.2
1	−x, −y, −z	19.01	0	−0.0	0	0	−0.0
2	x, y, z	6.65	−0.6	−1.1	−31.1	10.6	−20.8
1	−x, −y, −z	17.29	0	−0.0	0	0	−0.0
1	x, y, z	13.3	0	−0.0	0	0	−0.0
1	−x, −y, −z	18.03	0	−0.0	0	0	−0.0
1	-	6.83	0	0	0	0	0
1	-	10.66	−0.6	−1.1	−31.1	10.6	−20.8
1	-	10.45	0	−0.0	0	0	−0.0
1	-	20.71	0	0	0	0	0
1	-	7.64	−0.6	−1.1	−31.1	10.6	−20.8
1	-	18.77	0	−0.0	0	0	−0.0
1	-	5.99	1	−0.1	−1.9	0	−0.8
1	-	20.25	0	−0.0	0	0	−0.0
1	-	11.86	0	−0.0	0	0	−0.0
1	-	20.21	0	−0.0	0	0	−0.0
1	-	12.63	1.5	−0.1	−1.6	0	0
1	-	11.37	0	−0.0	0	0	−0.0
1	-	10.78	0	0	0	0	0
1	-	8.96	0	0	0	0	0
1	x, y, z	10.09	1	−0.1	−1.9	0	−0.8
1	−x, −y, −z	17.41	0	−0.0	0	0	−0.0
1	−x, −y, −z	20.96	0	−0.0	0	0	−0.0
1	-	8.53	0	−0.0	0	0	−0.0
1	-	12.56	0	−0.0	0	0	−0.0
1	-	9.32	−0.6	−1.1	−31.1	10.6	−20.8
1	-	19.08	0	−0.0	0	0	−0.0
1	-	4.41	1.5	−0.1	−1.6	0	0
1	-	22.25	0	−0.0	0	0	−0.0
1	-	11.36	0.6	−0.0	−0.3	0	0.3
1	-	11.9	0	−0.0	0	0	−0.0
1	-	7.35	−0.6	−1.1	−31.1	10.6	−20.8
1	-	10.73	0	−0.0	0	0	−0.0
1	-	10.02	−0.6	−1.1	−31.1	10.6	−20.8
1	-	8.56	0	−0.0	0	0	−0.0

**Table 7 ijms-24-05467-t007:** Interaction energies of the 3C17 molecular pairs involved in energy calculation in kJ/mol. R is the distance between molecular centroids in Å, and N is the number of molecular pairs involved.

N	Symop	R	E_ele	E_pol	E_dis	E_rep	E_tot
1	x, y, z	8.49	1.5	−0.1	−1.7	0	−0.1
1	−x, −y, −z	20.08	0	−0.3	0	0	−0.2
1	−x, −y, −z	20.18	0	−0.0	0	0	−0.0
2	x, y, z	6.69	−0.4	−1.1	−31.1	9.5	−21.4
1	−x, −y, −z	18.24	0	−0.0	0	0	−0.0
1	x, y, z	13.39	0	−0.0	0	0	−0.0
1	−x, −y, −z	18.66	0	−0.0	0	0	−0.0
1	-	7.38	0	0	0	0	0
1	-	11.08	−0.4	−1.1	−31.1	9.5	−21.4
1	-	11.1	0	−0.0	0	0	−0.0
1	-	22.01	0	−0.0	0	0	−0.0
1	-	7.7	−0.4	−1.1	−31.1	9.5	−21.4
1	-	19.89	0	−0.0	0	0	−0.0
1	-	5.99	1	−0.1	−2.0	0	−0.9
1	-	21.16	0	−0.0	0	0	−0.0
1	-	13.39	0	−0.0	0	0	−0.0
1	-	19.91	0	−0.0	0	0	−0.0
1	-	11.91	0	−0.0	0	0	−0.0
1	-	20.85	0	−0.0	0	0	−0.0
1	-	13.34	1.5	−0.1	−1.7	0	−0.1
1	-	11.62	0	−0.0	0	0	−0.0
1	-	11.33	0	−0.0	0	0	−0.0
1	-	11.2	0	0	0	0	0
1	x, y, z	10.15	1	−0.1	−2.0	0	−0.9
1	−x, −y, −z	18.52	0	−0.0	0	0	−0.0
1	−x, −y, −z	21.28	0	−0.0	0	0	−0.0
1	-	8.69	0	−0.0	0	0	−0.0
1	-	12.73	0	−0.0	0	0	−0.0
1	-	9.4	−0.4	−1.1	−31.1	9.5	−21.4
1	-	4.42	1.5	−0.1	−1.7	0	−0.1
1	-	22.06	0	−0.0	0	0	−0.0
1	-	22.61	0	−0.0	0	0	−0.0
1	-	11.9	0.6	−0.0	−0.3	0	0.3
1	-	11.92	0	−0.0	0	0	−0.0
1	-	7.71	−0.4	−1.1	−31.1	9.5	−21.4
1	-	11.02	0	−0.0	0	0	−0.0
1	-	8.76	−0.4	−1.1	−31.1	9.5	−21.4
1	-	10.42	0	−0.0	0	0	−0.0

**Table 8 ijms-24-05467-t008:** Calculation results of reactivity descriptors [51] of 3C16 and 3C17 molecules.

Descriptors	3C16, Values (eV)	3C17, Values (eV)
*E* _LUMO_	−2.6476	−2.7088
*E* _HOMO_	−2.9334	−2.9943
Energy gap (∆*E*)	0.2858	0.2855
Ionization energy (*I*)	2.9334	2.9943
Electron affinity (*A*)	2.6476	2.7088
Electronegativity (*χ*)	2.7905	2.8516
Chemical potential (*μ*)	−2.7905	−2.8516
Global hardness (*η*)	0.1429	0.1428
Global softness (*σ*)	3.4990	3.5014
Electrophilicity index (*ω*)	27.2459	28.4821

## Data Availability

The raw/processed data generated in this work are available upon request from the corresponding author.

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
