# Peer review of "A Comprehensive Investigation into the Crystallology, Molecule, and Quantum Chemistry Properties of Two New Hydrous Long-Chain Dibasic Ammonium Salts CnH2n+8N2O6 (n = 35 and 37)"

_ijms, 2023, doi:10.3390/ijms24065467_

Round 1

Reviewer 1 Report

1.      In section 2.4, kindly mention the version of Crystal Explorer.

2.      Table 1, WR2 value for compound 3C16 is really high. Kindly refine data. For a good crystal data and good refinement, WR2 value is less than 0.25. May be the data has twining.

3.      Kindly provide CIF for review.

4.      Kindly perform void analysis by using Crystal Explorer software. As the voids are the key factors to predict the mechanical properties of the single crystals. For help, see the following https://doi.org/10.1039/C0CE00683A; https://doi.org/10.1039/D2RA07681K; https://doi.org/10.1016/j.rechem.2022.100600

5.      Find the enrichment ratio for the pair of chemical species in compounds. For help see the following

https://doi.org/10.1107/S2052252514003327; https://doi.org/10.1021/acsomega.2c05441; https://doi.org/10.1016/j.molstruc.2022.133985

6.      Compare the crystal structures with the related crystal structures of the literature.

7.      Improve English as the manuscript contains some grammatical errors. 

Author Response

Dear editor,

The manuscript (ijms-2252641) entitled " Synthesis, Crystal structures, Hirshfeld surface analysis, and Quantum chemical caculations of two hydrous long-chain di-basic ammonium salts CnH2n+8N2O6 (n = 35 and 37)" has been revised mainly according to the reviewer 1’s comments, the style of the Journal and necessary scientific norms. The revisions made by us were explained as follows in order to facilitate your handling the manuscript:

1.In section 2.4, kindly mention the version of Crystal Explorer.

We have supplemented the version of Crystal Explorer in section 2.4 and marked it in red in the revised manuscript. The version of Crystal Explorer 17.5 is used in this paper.

  1. Table 1, wR2 value for compound 3C16 is really high. Kindly refine data. For a good crystal data and good refinement, wR2 value is less than 0.25. May be the data has twining.

Thank you very much for your constructive comments. There are many reasons for the high value of wR2, such as anisotropic processing, atom localization, hydrotreating, twinning, heteroatom interference, etc. We have checked these factors one by one and finally reduced the value of wR2 of 3C16 in Table 1 to 27.93%. At the same time, we also re-synthesized 3C16 samples in order to obtain more perfect crystals. For this kind of long chain single crystal, the period of sample synthesis needs at least four months. We did our best. I hope to get the understanding of reviewers and editors. Thanks again!

  1. Kindly provide CIF for review.

We have uploaded the CIF files of the samples to the submission system in the form of an attachment.

  1. Kindly perform void analysis by using Crystal Explorer software. As the voids are the key factors to predict the mechanical properties of the single crystals. For help, see the following:

https://doi.org/10.1107/S2052252514003327; https://doi.org/10.1021/acsomega.2c05441;

https://doi.org/10.1016/j.rechem.2022.100600.

According to the comment of the reviewer, we have carried out the void analysis of the two compounds by using Crystal Explorer software, as shown in the following figure (Table S1 was listed in attachment 1):

Table 1S. Void volumes, surface areas and void volumes as a percentage of total unit cell volume for different choices of isovalue for 3C16 and 3C17.

Molecule

Isovalue/au

Volume/Å3

Surface area/Å2

% of cell volume

3C16

0.0003

0.50

5.81

0.03

0.0005

10.44

64.82

0.53

0.0008

37.65

167.71

1.92

0.001

57.51

228.3

2.93

0.002

214.46

785.67

10.94

3C17

0.0003

0.17

2.09

0.01

0.0005

5.07

38.55

0.24

0.0008

30.3

157.41

1.45

0.001

52.49

250.69

2.52

0.002

248.87

912.08

11.93

Fig 9. Void surfaces (0.002 au) for the 3C16 and 3C17. The view is along the b axis and c axis in all cases, and all diagrams are on the same scale.

The relevant description was added to Section 3.2 of the revised manuscript and marked in red:

The mechanical strength of single crystal is related to the spatial crystal packing. The single crystals with large cavities show a limited capacity for withstanding external forces, whereas those without large cavities exhibit a notable ability to bear considerable forces or stresses [50, 51]. We carried out void analysis on 3C16 and 3C17 crystals, which is based on adding up the atomic electron density by using Hartree-Fock theory. It is assumed that all the atoms are spherically symmetric while calculating voids. When the electron density isosurface value is 0.002 au, the void volumes of 3C16 and 3C17 are 214.46 Å3 and 248.87 Å3, respectively. The volume of voids in 3C16 and 3C17 accounts for 10.94 % and 11.93 % of the total volume, respectively. Since the space occupied by the voids in the two compounds is very small, there is no large cavity in the crystal packing of 3C16 and 3C17. We can speculate that 3C16 and 3C17 have good mechanical properties.

Addding Reference:

  1. Ali, A.; Din, Z. U.; Ibrahim, M.; Ashfaq, M.; Muhammad, S.; Gull, D.; Tahir, M. N.; Rodrigues-Filho, E.; Al-Sehemi, A. G.; Suleman, M., Acid catalyzed one-pot approach towards the synthesis of curcuminoid systems: un-symmetrical diarylidene cycloalkanones, exploration of their single crystals, optical and nonlinear optical proper-ties. RSC Adv. 2023, 13, 4476-4494.
  2. Askerov, R. K.; Ashfaq, M.; Chipinsky, E. V.; Osmanov, V. K.; Tahir, M. N.; Baranov, E. V.; Fukin, G. K.; Khrus-talev, V. N.; Nazarov, R. H.; Borisova, G. N.; Matsulevich, Z. V.; Maharramov, A. M.; Borisov, A. V., Synthesis, crystal structure, exploration of the supramolecular assembly through Hirshfeld surface analysis and bactericidal activity of the cadmium organometallic complexes obtained from the heterocyclic ligand. Results in Chemistry 2022, 4, 100600.
  3. Find the enrichment ratio for the pair of chemical species in compounds. For help see the following:

https://doi.org/10.1107/S2052252514003327; https://doi.org/10.1021/acsomega.2c05441; https://doi.org/10.1016/j.molstruc.2022.133985

The three references of the reviewer had provided us with great help. Through Crystal Explorer 17.5  software and references, we calculated the enrichment rate of this pair of chemicals in the compound. (Table S2 and Table S3 were listed in attachment 1). The relevant description was added to Section 3.2 of the revised manuscript and marked in red:

The ability of a pair of chemical species (X, Y) to form crystal packing interactions is determined by computing the enrichment ratio. The enrichment ratio is calculated by dividing the proportion of the actual contacts by the theoretical proportion of the random contacts [52-54]. For a particular crystal, some contacts are more favorable to form crystal packing interactions than the other contacts. Enrichment ratio for a contact provides the tendency of it to form crystal packing interactions. The contacts with enrichment ratio greater than one have higher tendency to form crystal packing interactions as compare to other contacts. Table S2 and Table S3 list the enrichment ratios of all possible chemical pairs of 3C16 and C17. From Table S2, it can be seen that the enrichment ratios of C-H contact, O-H contact and H-H contact in 3C16 molecule are 0.83, 1.19 and 0.97. From Table S3, it can be seen that the enrichment ratios of C-H contact, O-H contact and H-H contact in 3C17 molecule are 0.89, 1.18 and 0.97. It can be seen that the O-H contact in the two molecules is beneficial contact.

Table S2. Hirshfeld contact surfaces and derived ‘random contact’ and ‘enrichment ratios’ for the 3C16 crystal.

Atoms

H

O

C

N

H

79.4

8.8

0.4

Contacts

O

10.6

0.2

_

(%)

C

0.5

_

_

_

N

0

_

_

_

Surface(%)

90.5

9.0

0.4

_

H

81.9

7.4

0.2

Contacts

O

8.9

0.8

_

Random

C

0.6

_

_

(%)

N

0

_

_

_

H

0.97

1.19

2.0

Enrichment

O

1.19

0.25

_

_

C

0.83

_

_

_

N

0

_

_

_

Table S3. Hirshfeld contact surfaces and derived ‘random contact’ and ‘enrichment ratios’ for the 3C17 crystal.

Atoms

H

O

C

N

H

80.0

8.5

0.4

Contacts

O

10.3

0.2

_

(%)

C

0.4

_

_

_

N

0

_

_

_

Surface(%)

90.7

8.8

0.5

_

H

82.3

7.2

0.45

Contacts

O

8.7

0.8

_

Random

C

0.45

_

_

(%)

N

0

_

_

_

H

0.97

1.18

0.89

Enrichment

O

1.18

0.25

_

_

C

0.89

_

_

_

N

_

_

_

_

Addding Reference:

  1. Jelsch, C.; Ejsmont, K.; Huder, L., The enrichment ratio of atomic contacts in crystals, an indicator derived from the Hirshfeld surface analysis. IUCrJ 2014, 1, 119-128.
  2. Ali, A.; Ashfaq, M.; Din, Z. U.; Ibrahim, M.; Khalid, M.; Assiri, M. A.; Riaz, A.; Tahir, M. N.; Rodrigues-Filho, E.; Imran, M.; Kuznetsov, A., Synthesis, Structural, and Intriguing Electronic Properties of Symmetrical Bis-Aryl-α,β-Unsaturated Ketone Derivatives. ACS Omega 2022, 7, 39294-39309.
  3. Faihan, A. S.; Aziz, N. M.; Ashfaq, M.; Hassan, W. M. I.; Al-Jibori, S. A.; Al-Janabi, A. S.; Tahir, M. N.; Al-barwari, A. S. M. O., Synthesis, characterization, and x-ray crystallography of unexpected chloro-substitution on 1-(4-chlorophenyl)-3-phenylthiourea platinum(II) complex with tertiary phosphine ligand. J. Mol. Struct. 2022, 1270, 133985.
  4. Compare the crystal structures with the related crystal structures of the literature.

We have searched the relevant literature and compared the crystal structure with the relevant crystal structure in the literature. The supplementary content is located in section 3.1 of the revised manuscript and marked in red. The supplementary contents and references are as follows:

The two crystal structures have the same crystal system and space group as reported in the literature [41]. Unlike the compound reported in the literature, the number of mole-cules in a single crystal cell and the length of molecules are different.

  1. Zhang, L.-J.; Di, Y.-Y.; Dou, J.-M., Crystal structure and standard molar enthalpy of formation of ethylenedia-mine dilauroleate (C12H24O2)2C2N2H8(s). J. Therm. Anal. Calorim. 2013, 114, 359-363.
  2. Improve English as the manuscript contains some grammatical errors.

We have revised the grammatical errors in the full text and marked them in red in the revised manuscript.

We have resubmitted the manuscript to the editorial office of the International Journal of Molecular Sciences and we are very grateful to you for kind considering publication of the manuscript in your journal.

With my best regards,

Sincerely Yours,

Prof. Xin-Hui Fan

Dear editor,

The manuscript (ijms-2252641) entitled " Synthesis, Crystal structures, Hirshfeld surface analysis, and Quantum chemical caculations of two hydrous long-chain di-basic ammonium salts CnH2n+8N2O6 (n = 35 and 37)" has been revised mainly according to the reviewer 1’s comments, the style of the Journal and necessary scientific norms. The revisions made by us were explained as follows in order to facilitate your handling the manuscript:

1.In section 2.4, kindly mention the version of Crystal Explorer.

We have supplemented the version of Crystal Explorer in section 2.4 and marked it in red in the revised manuscript. The version of Crystal Explorer 17.5 is used in this paper.

  1. Table 1, wR2 value for compound 3C16 is really high. Kindly refine data. For a good crystal data and good refinement, wR2 value is less than 0.25. May be the data has twining.

Thank you very much for your constructive comments. There are many reasons for the high value of wR2, such as anisotropic processing, atom localization, hydrotreating, twinning, heteroatom interference, etc. We have checked these factors one by one and finally reduced the value of wR2 of 3C16 in Table 1 to 27.93%. At the same time, we also re-synthesized 3C16 samples in order to obtain more perfect crystals. For this kind of long chain single crystal, the period of sample synthesis needs at least four months. We did our best. I hope to get the understanding of reviewers and editors. Thanks again!

  1. Kindly provide CIF for review.

We have uploaded the CIF files of the samples to the submission system in the form of an attachment.

  1. Kindly perform void analysis by using Crystal Explorer software. As the voids are the key factors to predict the mechanical properties of the single crystals. For help, see the following:

https://doi.org/10.1107/S2052252514003327; https://doi.org/10.1021/acsomega.2c05441;

https://doi.org/10.1016/j.rechem.2022.100600.

According to the comment of the reviewer, we have carried out the void analysis of the two compounds by using Crystal Explorer software, as shown in the following figure (Table S1 was listed in attachment 1):

Table 1S. Void volumes, surface areas and void volumes as a percentage of total unit cell volume for different choices of isovalue for 3C16 and 3C17.

Molecule

Isovalue/au

Volume/Å3

Surface area/Å2

% of cell volume

3C16

0.0003

0.50

5.81

0.03

0.0005

10.44

64.82

0.53

0.0008

37.65

167.71

1.92

0.001

57.51

228.3

2.93

0.002

214.46

785.67

10.94

3C17

0.0003

0.17

2.09

0.01

0.0005

5.07

38.55

0.24

0.0008

30.3

157.41

1.45

0.001

52.49

250.69

2.52

0.002

248.87

912.08

11.93

Fig 9. Void surfaces (0.002 au) for the 3C16 and 3C17. The view is along the b axis and c axis in all cases, and all diagrams are on the same scale.

The relevant description was added to Section 3.2 of the revised manuscript and marked in red:

The mechanical strength of single crystal is related to the spatial crystal packing. The single crystals with large cavities show a limited capacity for withstanding external forces, whereas those without large cavities exhibit a notable ability to bear considerable forces or stresses [50, 51]. We carried out void analysis on 3C16 and 3C17 crystals, which is based on adding up the atomic electron density by using Hartree-Fock theory. It is assumed that all the atoms are spherically symmetric while calculating voids. When the electron density isosurface value is 0.002 au, the void volumes of 3C16 and 3C17 are 214.46 Å3 and 248.87 Å3, respectively. The volume of voids in 3C16 and 3C17 accounts for 10.94 % and 11.93 % of the total volume, respectively. Since the space occupied by the voids in the two compounds is very small, there is no large cavity in the crystal packing of 3C16 and 3C17. We can speculate that 3C16 and 3C17 have good mechanical properties.

Addding Reference:

  1. Ali, A.; Din, Z. U.; Ibrahim, M.; Ashfaq, M.; Muhammad, S.; Gull, D.; Tahir, M. N.; Rodrigues-Filho, E.; Al-Sehemi, A. G.; Suleman, M., Acid catalyzed one-pot approach towards the synthesis of curcuminoid systems: un-symmetrical diarylidene cycloalkanones, exploration of their single crystals, optical and nonlinear optical proper-ties. RSC Adv. 2023, 13, 4476-4494.
  2. Askerov, R. K.; Ashfaq, M.; Chipinsky, E. V.; Osmanov, V. K.; Tahir, M. N.; Baranov, E. V.; Fukin, G. K.; Khrus-talev, V. N.; Nazarov, R. H.; Borisova, G. N.; Matsulevich, Z. V.; Maharramov, A. M.; Borisov, A. V., Synthesis, crystal structure, exploration of the supramolecular assembly through Hirshfeld surface analysis and bactericidal activity of the cadmium organometallic complexes obtained from the heterocyclic ligand. Results in Chemistry 2022, 4, 100600.
  3. Find the enrichment ratio for the pair of chemical species in compounds. For help see the following:

https://doi.org/10.1107/S2052252514003327; https://doi.org/10.1021/acsomega.2c05441; https://doi.org/10.1016/j.molstruc.2022.133985

The three references of the reviewer had provided us with great help. Through Crystal Explorer 17.5  software and references, we calculated the enrichment rate of this pair of chemicals in the compound. (Table S2 and Table S3 were listed in attachment 1). The relevant description was added to Section 3.2 of the revised manuscript and marked in red:

The ability of a pair of chemical species (X, Y) to form crystal packing interactions is determined by computing the enrichment ratio. The enrichment ratio is calculated by dividing the proportion of the actual contacts by the theoretical proportion of the random contacts [52-54]. For a particular crystal, some contacts are more favorable to form crystal packing interactions than the other contacts. Enrichment ratio for a contact provides the tendency of it to form crystal packing interactions. The contacts with enrichment ratio greater than one have higher tendency to form crystal packing interactions as compare to other contacts. Table S2 and Table S3 list the enrichment ratios of all possible chemical pairs of 3C16 and C17. From Table S2, it can be seen that the enrichment ratios of C-H contact, O-H contact and H-H contact in 3C16 molecule are 0.83, 1.19 and 0.97. From Table S3, it can be seen that the enrichment ratios of C-H contact, O-H contact and H-H contact in 3C17 molecule are 0.89, 1.18 and 0.97. It can be seen that the O-H contact in the two molecules is beneficial contact.

Table S2. Hirshfeld contact surfaces and derived ‘random contact’ and ‘enrichment ratios’ for the 3C16 crystal.

Atoms

H

O

C

N

H

79.4

8.8

0.4

Contacts

O

10.6

0.2

_

(%)

C

0.5

_

_

_

N

0

_

_

_

Surface(%)

90.5

9.0

0.4

_

H

81.9

7.4

0.2

Contacts

O

8.9

0.8

_

Random

C

0.6

_

_

(%)

N

0

_

_

_

H

0.97

1.19

2.0

Enrichment

O

1.19

0.25

_

_

C

0.83

_

_

_

N

0

_

_

_

Table S3. Hirshfeld contact surfaces and derived ‘random contact’ and ‘enrichment ratios’ for the 3C17 crystal.

Atoms

H

O

C

N

H

80.0

8.5

0.4

Contacts

O

10.3

0.2

_

(%)

C

0.4

_

_

_

N

0

_

_

_

Surface(%)

90.7

8.8

0.5

_

H

82.3

7.2

0.45

Contacts

O

8.7

0.8

_

Random

C

0.45

_

_

(%)

N

0

_

_

_

H

0.97

1.18

0.89

Enrichment

O

1.18

0.25

_

_

C

0.89

_

_

_

N

_

_

_

_

Addding Reference:

  1. Jelsch, C.; Ejsmont, K.; Huder, L., The enrichment ratio of atomic contacts in crystals, an indicator derived from the Hirshfeld surface analysis. IUCrJ 2014, 1, 119-128.
  2. Ali, A.; Ashfaq, M.; Din, Z. U.; Ibrahim, M.; Khalid, M.; Assiri, M. A.; Riaz, A.; Tahir, M. N.; Rodrigues-Filho, E.; Imran, M.; Kuznetsov, A., Synthesis, Structural, and Intriguing Electronic Properties of Symmetrical Bis-Aryl-α,β-Unsaturated Ketone Derivatives. ACS Omega 2022, 7, 39294-39309.
  3. Faihan, A. S.; Aziz, N. M.; Ashfaq, M.; Hassan, W. M. I.; Al-Jibori, S. A.; Al-Janabi, A. S.; Tahir, M. N.; Al-barwari, A. S. M. O., Synthesis, characterization, and x-ray crystallography of unexpected chloro-substitution on 1-(4-chlorophenyl)-3-phenylthiourea platinum(II) complex with tertiary phosphine ligand. J. Mol. Struct. 2022, 1270, 133985.
  4. Compare the crystal structures with the related crystal structures of the literature.

We have searched the relevant literature and compared the crystal structure with the relevant crystal structure in the literature. The supplementary content is located in section 3.1 of the revised manuscript and marked in red. The supplementary contents and references are as follows:

The two crystal structures have the same crystal system and space group as reported in the literature [41]. Unlike the compound reported in the literature, the number of mole-cules in a single crystal cell and the length of molecules are different.

  1. Zhang, L.-J.; Di, Y.-Y.; Dou, J.-M., Crystal structure and standard molar enthalpy of formation of ethylenedia-mine dilauroleate (C12H24O2)2C2N2H8(s). J. Therm. Anal. Calorim. 2013, 114, 359-363.
  2. Improve English as the manuscript contains some grammatical errors.

We have revised the grammatical errors in the full text and marked them in red in the revised manuscript.

We have resubmitted the manuscript to the editorial office of the International Journal of Molecular Sciences and we are very grateful to you for kind considering publication of the manuscript in your journal.

With my best regards,

Sincerely Yours,

Prof. Xin-Hui Fan

Reviewer 2 Report

- You may change the phrase: "the two title compounds..." used in several parts.

Author Response

According to the style of the International Journal of Molecular Science, we have changed "the two title compounds" to "the two molecules" and marked it in red in the revised manuscript.

Reviewer 3 Report

Reviewer’s Comments:

The manuscript “Synthesis, Crystal structures, Hirshfeld surface analysis, and Quantum chemical caculations of two hydrous long-chain dibasic ammonium salts CnH2n+8N2O6 (n = 35 and 37)” is a very interesting work. In this work, through the salification reaction of carboxylation, successful attachment of long-chain alkanoic acid to the two ends of 1, 3-propanediamine was realized, which enabled the doubling of long-chain alkanoic acid carbon chain. Hydrous 1, 3-propanediamine dihexadecanoate (abbreviated as 3C16) and 1, 3-propanediamine diheptadecanoate (abbreviated as 3C17) were synthesized afterwards and their crystal structures were characterized by X-ray single crystal diffraction technique. By analyzing their molecular and crystal structure, composition, spatial structure, and coordination mode were determined. Two water molecules played important roles in stabilizing the framework of both compounds. While I believe this topic is of great interest to our readers, I think it needs major revision before it is ready for publication. So, I recommend this manuscript for publication with major revisions.

1. In this manuscript, the authors did not explain the importance of the dibasic ammonium salts the introduction part. The authors should explain the importance of dibasic ammonium salts.

2) Title: The title of the manuscript is not impressive. It should be modified or rewritten it.

3) Correct the following statement “Calculations with DFT unveiled the energies of front orbit HOMO and LUMO of the two title compounds, with gap values of 0.2858 eV and 0.2855 eV, respectively. Using the multiwfn package and the VMD visualization software, the density of states (DOS) and the electrostatic potential (ESP) maps of the two title compounds were obtained”.

4) Keywords: The dibasic ammonium salts is missing in the keywords. So, modify the keywords.

5) Introduction part is not impressive. The references cited are very old. So, Improve it with some latest literature like 10.3390/molecules27217368, 10.3390/molecules27196580

6) The authors should explain the following statement with recent references, “Hydrophilic groups of both compounds are located inside the cell. This can also be clearly seen from the 2D space stacking diagrams in Fig. 5 and the 3D space filling diagram in Fig. 6”.

7) Add space between magnitude and unit. For example, in synthesis “21.96g” should be 21.96 g. Make the corrections throughout the manuscript regarding values and units.

8) The author should provide reason about this statement “The positive electro-positive sites of 3C16 are dispersed around amine, with the maximum electrostatic potential of 110.44 kcal/mol”.

9. Comparison of the present results with other similar findings in the literature should be discussed in more detail. This is necessary in order to place this work together with other work in the field and to give more credibility to the present results.

10) Conclusion part is very long. Make it brief and improve by adding the results of your studies.

11) There are many grammatic mistakes. Improve the English grammar of the manuscript.

Author Response

Dear editor,

The manuscript (ijms-2252641) entitled " Synthesis, Crystal structures, Hirshfeld surface analysis, and Quantum chemical caculations of two hydrous long-chain di-basic ammonium salts CnH2n+8N2O6 (n = 35 and 37)" has been revised mainly according to the reviewer 1’s comments, the style of the Journal and necessary scientific norms. The revisions made by us were explained as follows in order to facilitate your handling the manuscript:

The manuscript “Synthesis, Crystal structures, Hirshfeld surface analysis, and Quantum chemical caculations of two hydrous long-chain dibasic ammonium salts CnH2n+8N2O6 (n = 35 and 37)” is a very interesting work. In this work, through the salification reaction of carboxylation, successful attachment of long-chain alkanoic acid to the two ends of 1, 3-propanediamine was realized, which enabled the doubling of long-chain alkanoic acid carbon chain. Hydrous 1, 3-propanediamine dihexadecanoate (abbreviated as 3C16) and 1, 3-propanediamine diheptadecanoate (abbreviated as 3C17) were synthesized afterwards and their crystal structures were characterized by X-ray single crystal diffraction technique. By analyzing their molecular and crystal structure, composition, spatial structure, and coordination mode were determined. Two water molecules played important roles in stabilizing the framework of both compounds. While I believe this topic is of great interest to our readers, I think it needs major revision before it is ready for publication. So, I recommend this manuscript for publication with major revisions.

  • In this manuscript, the authors did not explain the importance of the dibasic ammonium salts the introduction part. The authors should explain the importance of dibasic ammonium salts.

This constructive opinion play a huge role in improving the overall framework of the manuscript. We sincerely thank the reviewer. According to the suggestion of the reviewer, we have made a supplement to the "Introduction" section, and the supplementary contents are as follows:

As a nucleophilic reagent, 1,3-propylene diamine is alkaline and can form hydrogen bonds. It is often used as an intermediate and solvent in organic synthesis [35, 36]. In addition, 1,3-propanediamine plays an important role in photosynthesis and the cultivation of biological strains [37, 38]. Amines and their derivatives have been widely reported [39, 40]. The binary ammonium salt formed with ethylenediamine and lauric acid as ligands had been reported in the literature [41]. The results showed that this kind of binary ammonium salt had good thermodynamic properties. However, the synthesis of long-chain binary ammonium salts with 1,3-propanediamine and long-chain fatty acids as ligands has not been reported. The study of binary ammonium salt by quantum chemical calculation [42-45] has never been reported.

2) Title: The title of the manuscript is not impressive. It should be modified or rewritten it.

We have revised the title of this paper and marked it in red in the revised manuscript. The revised title is as follows:

A Comprehensive Investigation into the Crystallology, Molecule and Quantum Chemistry Properties of Two New Hydrous Long-chain Dibasic Ammonium Salts CnH2n+8N2O6 (n=35 and 37)

3) Correct the following statement “Calculations with DFT unveiled the energies of front orbit HOMO and LUMO of the two title compounds, with gap values of 0.2858 eV and 0.2855 eV, respectively. Using the multiwfn package and the VMD visualization software, the density of states (DOS) and the electrostatic potential (ESP) maps of the two title compounds were obtained”.

We have revised this sentence and marked it in red in the revised manuscript. Modify as follows:

The DFT calculations were performed to analyze the frontier molecular orbitals (HOMO - LUMO). The energy difference between HOMO - LUMO is 0.2858 eV and 0.2855 eV of 3C16 and 3C17, respectively. DOS diagrams further confirmed the distribution of frontier molecular orbitals of 3C16 and 3C17. The charge distributions in the compounds were visualised using molecular electrostatic potential (ESP) surface. ESP maps indicated that the electrophilic sites are localized around the oxygen atom.

4) Keywords: The dibasic ammonium salts is missing in the keywords. So, modify the keywords.

We have added the keywords mentioned by the reviewer to the revised manuscript.

5) Introduction part is not impressive. The references cited are very old. So, Improve it with some latest literature like 10.3390/molecules27217368, 10.3390/molecules27196580

We have improved the "introduction" section, and see the first reply for the supplementary content. At the same time, we have adjusted the references and cited the latest references, including the two documents mentioned by the reviewers. The newly added references are listed as follows:

  1. Nazarychev, V. M.; Glova, A. D.; Larin, S. V.; Lyulin, A. V.; Lyulin, S. V.; Gurtovenko, A. A., Cooling-Rate Computer Simulations for the Description of Crystallization of Organic Phase-Change Materials. Int. J. Mol. Sci. 2022, 23, 14567-14582.
  2. Topor, A.; Liu, D.; Maxim, C.; Novitchi, G.; Train, C.; AlOthman, Z. A.; Al-Kahtani, A. A. S.; Ungur, L.; Ho, L. T. A.; Chibotaru, L. F.; Andruh, M., Design of FeIII–LnIII binuclear complexes using compartmental ligands: synthesis, crystal structures, magnetic properties, and ab initio analysis. J. Mater. Chem. C 2021, 9, 10912-10926.
  3. Ni, G.-H.; Sun, Y.-X.; Ji, C.-X.; Jin, Y.-W.; Liu, M.-Y.; Zhao, J.-P.; Liu, F.-C., Plasticity and Ferroelasticity Tran-sitions of Molecular Complex [(C4H9N2)2][Fe3O(O2CH)9] on Heating and Cooling near Room Temperature. Cryst. Growth Des. 2022, 22, 3428-3434.
  4. Zhgun, A. A.; Eldarov, M. A., Spermidine and 1,3-Diaminopropane Have Opposite Effects on the Final Stage of Cephalosporin C Biosynthesis in High-Yielding Acremonium chrysogenum Strain. Int. J. Mol. Sci. 2022, 23, 14625-14648.
  5. Gondor, O. K.; Tajti, J.; Hamow, K. Á.; Majláth, I.; Szalai, G.; Janda, T.; Pál, M., Polyamine Metabolism under Different Light Regimes in Wheat. Int. J. Mol. Sci. 2021, 22, 11717-11743.
  6. Mumtaz, S.; Iqbal, S.; Shah, M.; Hussain, R.; Rahim, F.; Rehman, W.; Khan, S.; Abid, O.-u.-R.; Rasheed, L.; Dera, A. A.; Al-ghulikah, H. A.; Kehili, S.; Elkaeed, E. B.; Alrbyawi, H.; Alahmdi, M. I., New Triazinoindole Bearing Benzimidazole/Benzoxazole Hybrids Analogs as Potent Inhibitors of Urease: Synthesis, In Vitro Analysis and Molecular Docking Studies. Molecules 2022, 27, 6580-6598.
  7. Khan, S.; Iqbal, S.; Taha, M.; Rahim, F.; Shah, M.; Ullah, H.; Bahadur, A.; Alrbyawi, H.; Dera, A. A.; Alahmdi, M. I.; Pashameah, R. A.; Alzahrani, E.; Farouk, A.-E., Synthesis, In Vitro Biological Evaluation and In Silico Molecular Docking Studies of Indole Based Thiadiazole Derivatives as Dual Inhibitor of Acetylcholinesterase and Butyrylchloinesterase. Molecules 2022, 27, 7368-7379.
  8. Zhang, L.-J.; Di, Y.-Y.; Dou, J.-M., Crystal structure and standard molar enthalpy of formation of ethylenedia-mine dilauroleate (C12H24O2)2C2N2H8(s). J. Therm. Anal. Calorim. 2013, 114, 359-363.
  9. Khan, S.; Ullah, H.; Taha, M.; Rahim, F.; Sarfraz, M.; Iqbal, R.; Iqbal, N.; Hussain, R.; Ali Shah, S. A.; Ayub, K.; Albalawi, M. A.; Abdelaziz, M. A.; Alatawi, F. S.; Khan, K. M., Synthesis, DFT Studies, Molecular Docking and Biological Activity Evaluation of Thiazole-Sulfonamide Derivatives as Potent Alzheimer’s Inhibitors. Molecules 2023, 28, 559-585.
  10. Alghuwainem, Y. A. A.; El-Lateef, H. M. A.; Khalaf, M. M.; Amer, A. A.; Abdelhamid, A. A.; Alzharani, A. A.; Alfarsi, A.; Shaaban, S.; Gouda, M.; Abdou, A., Synthesis, DFT, Biological and Molecular Docking Analysis of Novel Manganese(II), Iron(III), Cobalt(II), Nickel(II), and Copper(II) Chelate Complexes Ligated by 1-(4-Nitrophenylazo)-2-naphthol. Int. J. Mol. Sci. 2022, 23, 15614-15633.
  11. Bykov, A. V.; Shestimerova, T. A.; Bykov, M. A.; Osminkina, L. A.; Kuznetsov, A. N.; Gontcharenko, V. E.; Shevelkov, A. V., Synthesis, Crystal, and Electronic Structure of (HpipeH2)2[Sb2I10](I2), with I2 Molecules Link-ing Sb2X10 Dimers into a Polymeric Anion: A Strategy for Optimizing a Hybrid Compound’s Band Gap. Int. J. Mol. Sci. 2022, 24, 2201-2215.
  12. Li, L.; Zhang, Q.; Wei, Y.; Wang, Q.; Wang, W., Theoretical Study on the Gas Phase and Gas–Liquid Inter-face Reaction Mechanism of Criegee Intermediates with Glycolic Acid Sulfate. Int. J. Mol. Sci. 2023, 24, 3355-3365.

6) The authors should explain the following statement with recent references, “Hydrophilic groups of both compounds are located inside the cell. This can also be clearly seen from the 2D space stacking diagrams in Fig. 5 and the 3D space filling diagram in Fig. 6”.

In fact, the work of this paper is the continuation of the previous work. As early as ten years ago, our research group has prepared corresponding binary alkanes with ethylenediamine and lauric acid as ligands. ccording to the comments of the reviewer, we have quoted the relevant documents previously published in the revised manuscript. The description of organic and inorganic layers of sample molecules can be found in this reference. As follows:

DOI:10.1007/s10973-012-2929-7.

  1. Zhang, L.-J.; Di, Y.-Y.; Dou, J.-M., Crystal structure and standard molar enthalpy of formation of ethylenedia-mine dilauroleate (C12H24O2)2C2N2H8(s). J. Therm. Anal. Calorim. 2013, 114, 359-363.

7) Add space between magnitude and unit. For example, in synthesis “21.96g” should be 21.96 g. Make the corrections throughout the manuscript regarding values and units.

According to the suggestions of the reviewer, we have checked and revised the whole manuscript in detail. Revisions are marked in red in the revised manuscript.

8) The author should provide reason about this statement “The positive electro-positive sites of 3C16 are dispersed around amine, with the maximum electrostatic potential of 110.44 kcal/mol”.

Thank you for your constructive suggestion. It is easy to obtain the distribution of molecular surface electrostatic potential of 3C16 and C17 through the calculation of Multiwfn program package. The detailed information is listed in Fig. S1, Table S1 and Table S2. (Fig. S1, Table S4 and Table S5 are listed in attachment 1). From this information, we can easily find the extreme point of surface electrostatic potential. The added content has been marked in red in section 4.1.4 of the revised manuscript.

Table S4. The electrostatic potential distribution on the molecular surface of 3C16 (the data with * indicates the extreme point).

Number of surface minima

a.u.

eV

kcal/mol

X/Y/Z coordinate(Angstrom)

1

-0.16453

-4.47702

-103.243

-11.2868

-2.02493

-0.23548

2

-0.17027

-4.63317

-106.844

-8.72145

-1.60206

-1.2858

*3

-0.18087

-4.92175

-113.498

-7.75474

-2.55052

-2.37115

4

-0.02438

-0.66343

-15.299

-6.35988

3.411512

-1.94315

5

0.154834

4.213252

97.15999

-4.65438

7.243617

4.013048

6

-0.02839

-0.77248

-17.8137

-3.8894

4.060429

-5.12827

7

0.131806

3.586613

82.70933

-1.67138

7.080439

2.231581

8

0.008044

0.218889

5.047709

-1.29557

0.681204

1.492379

9

0.005605

0.152513

3.517026

0.000434

0.001186

-0.60823

10

0.013988

0.380633

8.777612

1.270841

1.117332

1.878849

11

0.013392

0.364421

8.403752

1.533637

0.31979

0.159183

12

0.005996

0.163173

3.762859

2.637313

0.520624

-2.45775

13

-0.0015

-0.04087

-0.9424

3.929375

-2.81683

2.71099

14

0.013188

0.358868

8.275706

3.939541

1.602784

2.050828

15

0.015495

0.421627

9.722963

4.078786

0.770223

0.428281

16

0.007731

0.210363

4.851096

5.118993

1.002794

-2.08478

17

-0.00279

-0.07603

-1.75324

5.270422

-3.3381

-1.28522

18

0.001694

0.046096

1.062991

6.761744

-2.35995

2.986289

19

0.010461

0.284664

6.564516

7.662142

2.63332

2.302381

20

-0.00043

-0.01171

-0.27009

8.177383

-2.94931

-1.1422

21

0.00272

0.07402

1.706946

10.46661

-1.22758

3.099985

22

0.000575

0.015647

0.360817

10.79381

-2.45708

-0.8996

23

0.004376

0.119074

2.745905

10.71045

2.33813

-1.40719

24

-0.00049

-0.01324

-0.30526

14.46594

-1.02014

-0.51419

Number of surface maxima

a.u.

eV

kcal/mol

X/Y/Zcoordinate(Angstrom)

1

0.174084

4.737079

109.2397

-6.24712

8.491867

1.227791

2

-0.0646

-1.75798

-40.54

-4.96487

-2.28594

-0.26814

3

-0.00379

-0.10307

-2.37695

-4.37301

0.499905

2.836129

4

0.163501

4.449091

102.5986

-3.95133

8.562197

-0.84213

5

0.168485

4.584708

105.726

-3.54038

6.048605

2.611593

6

0.014768

0.401862

9.267171

-2.17544

0.961225

2.88742

7

0.009981

0.271603

6.263316

-1.90906

0.695334

0.769408

8

0.171845

4.676127

107.8341

-1.11462

5.5864

0.633291

*9

0.176002

4.789245

110.4427

-1.01468

7.709102

-2.40349

10

0.007469

0.20325

4.68706

-0.7266

0.301428

-0.90077

11

0.02187

0.59511

13.72357

-0.00455

1.318212

3.097962

12

0.020062

0.545926

12.58937

0.249353

1.020814

0.699309

13

0.016749

0.455752

10.5099

1.397522

0.714272

-0.63209

14

0.008902

0.242232

5.586006

1.878454

0.35995

-3.47918

15

0.022175

0.603413

13.91506

2.339536

1.63042

3.454155

16

0.02077

0.565185

13.0335

2.544838

1.286011

0.626853

17

0.017867

0.486175

11.21146

3.79263

1.090075

-0.38668

18

0.013111

0.356769

8.227294

4.20413

0.624539

-3.5348

19

0.020176

0.54902

12.66071

4.763124

1.869393

3.927882

20

0.018522

0.504011

11.62277

5.015043

1.548659

0.572946

21

0.016798

0.457102

10.54103

6.177074

1.367294

0.016231

22

0.014252

0.387822

8.9434

6.696013

0.756833

-3.57825

23

0.018569

0.505275

11.65193

7.137496

2.261866

4.170209

24

0.016862

0.458836

10.58101

7.487066

1.947482

0.644759

25

0.010723

0.291781

6.728637

8.569841

-1.60058

-3.35366

26

0.015556

0.4233

9.761539

8.838187

-0.55271

4.94698

27

0.013597

0.369987

8.532115

9.128449

0.916611

-3.49573

28

0.013782

0.37502

8.64817

9.206488

-1.54296

1.176996

29

0.01823

0.496064

11.43952

10.31732

2.08734

2.568623

30

0.011121

0.302609

6.978333

10.68749

-0.03747

-3.00227

31

0.01146

0.311839

7.191191

11.21573

-1.20705

-3.09451

32

0.014313

0.389483

8.98171

11.19676

1.932719

0.807084

33

0.013069

0.355635

8.201142

11.76762

1.432635

-3.22327

34

0.012543

0.341305

7.870698

12.33976

-0.36814

-2.74104

35

0.013009

0.35398

8.162988

12.53867

-0.46368

1.305257

36

0.012624

0.343527

7.921918

12.91783

-0.97854

-2.75194

37

0.015191

0.413358

9.532272

13.98223

2.190865

-0.89459

Table S5. The electrostatic potential distribution on the molecular surface of 3C17 (the data with * indicates the extreme point).

Number of surface minima

a.u.

eV

kcal/mol

X/Y/Z coordinate(Angstrom)

1

-0.1636

-4.45181

-102.661

-11.9085

-1.99734

-0.24348

2

-0.1685

-4.58513

-105.736

-9.40271

-1.59916

-1.29443

*3

-0.17935

-4.88029

-112.542

-8.38295

-2.55955

-2.39748

4

-0.02899

-0.78883

-18.1908

-6.79626

3.392991

-2.03949

5

-0.03213

-0.87436

-20.1633

-4.41309

3.92474

-5.06832

6

0.132711

3.611239

83.27722

-2.19178

6.814799

2.19633

7

0.007183

0.195463

4.507477

-1.85527

0.651875

1.34083

8

0.013252

0.360613

8.315941

0.79948

1.058245

1.744755

9

0.012426

0.338122

7.797281

1.04878

0.244777

0.143264

10

0.005214

0.141886

3.271971

2.075822

0.460598

-2.52293

11

-0.00153

-0.04151

-0.95732

3.349939

-2.88238

2.710862

12

0.012335

0.335653

7.740345

3.41711

1.473553

1.984267

13

0.014612

0.397623

9.169403

3.567028

0.785782

0.521668

14

0.007135

0.194164

4.477527

4.62557

0.913192

-2.23322

15

-0.00323

-0.08788

-2.02662

4.720756

-3.37861

-1.33302

16

0.001021

0.027785

0.640734

6.195344

-2.54961

2.910019

17

-0.0003

-0.00818

-0.18872

7.630265

-3.00132

-1.07379

18

0.008825

0.240127

5.537476

7.654374

2.347407

2.249425

19

0.002344

0.063774

1.470674

8.812201

-2.19961

3.029691

20

0.005134

0.13971

3.221789

9.359405

1.814437

-1.83944

21

0.000274

0.007457

0.171956

10.41686

-2.5685

-0.93143

22

0.003978

0.108234

2.495947

10.94275

2.321264

-1.72689

23

0.004883

0.132862

3.063862

11.38995

1.837437

2.798236

24

0.00035

0.009532

0.219819

12.52812

-2.18248

-0.67011

25

0.001407

0.038292

0.88303

15.11443

2.002805

-1.16266

Number of surface maxima

a.u.

eV

kcal/mol

X/Y/Z coordinate(Angstrom)

*1

0.176608

4.805753

110.8234

-6.76955

8.628194

1.322795

2

-0.06353

-1.72885

-39.8682

-5.55241

-2.26965

-0.25296

3

-0.00364

-0.09907

-2.28449

-4.99689

0.544983

2.830868

4

0.163558

4.450638

102.6342

-4.38887

8.69457

-0.79437

5

0.170789

4.647402

107.1717

-4.08265

6.180803

2.686758

6

0.014863

0.40445

9.326856

-2.69532

0.977395

2.901049

7

0.008834

0.240387

5.543464

-2.43156

0.69611

0.793618

8

0.170751

4.646375

107.148

-1.63276

5.651626

0.61141

9

0.174342

4.744086

109.4013

-1.63689

7.709999

-2.51374

10

0.006046

0.164511

3.793715

-1.22933

0.221964

-0.80932

11

0.021766

0.592286

13.65846

-0.51993

1.327431

3.065997

12

0.019051

0.518403

11.95468

-0.30384

0.989092

0.661324

13

0.015987

0.435018

10.03176

0.935921

0.66741

-0.63873

14

0.007909

0.215207

4.962794

1.408729

0.297469

-3.50218

15

0.021901

0.595958

13.74313

1.851407

1.5785

3.453887

16

0.019796

0.53869

12.4225

2.083969

1.173466

0.506253

17

0.017339

0.471827

10.8806

3.295408

1.019625

-0.40342

18

0.012472

0.339367

7.826008

3.705203

0.492148

-3.65773

19

0.0198

0.538797

12.42496

4.34317

1.746123

3.899713

20

0.017859

0.48597

11.20674

4.626758

1.450578

0.520384

21

0.016343

0.444719

10.25548

5.684332

1.302464

-0.12283

22

0.013652

0.371493

8.566835

6.071878

-0.56799

4.761854

23

0.013514

0.367724

8.479921

6.196842

0.638312

-3.71757

24

0.017652

0.480348

11.07711

6.886938

2.002906

4.156496

25

0.015739

0.428274

9.87624

7.127111

1.673343

0.546919

26

0.011124

0.30271

6.980666

7.79932

-1.88915

0.829915

27

0.010453

0.284429

6.559105

8.077398

-1.99989

-3.40163

28

0.015267

0.415426

9.579951

8.311999

1.615191

0.196835

29

0.013359

0.363512

8.382801

8.484407

-0.6657

4.946163

30

0.013244

0.360375

8.310448

8.718766

0.760336

-3.68674

31

0.011355

0.30899

7.125477

9.001618

-1.83473

0.980333

32

0.017944

0.488273

11.25985

9.238059

2.406352

4.163059

33

0.016078

0.437508

10.08918

9.627346

1.840446

0.643807

34

0.011265

0.306531

7.068774

10.15856

-1.34879

1.054332

35

0.010909

0.296838

6.845255

10.41396

-1.65276

-3.23827

36

0.012608

0.343073

7.911455

11.39364

1.055675

-3.5221

37

0.015235

0.414555

9.559876

11.73779

-0.41798

3.296805

38

0.011526

0.313649

7.232919

12.41312

-0.59844

1.258983

39

0.010665

0.290214

6.6925

12.92971

-1.37426

-2.92657

40

0.014306

0.389288

8.977193

13.09965

2.255691

0.572747

41

0.013745

0.374019

8.62508

13.57756

1.403746

-3.22823

42

0.013283

0.361453

8.335319

15.52388

-0.56621

-0.42381

Figure S1. Molecular surface electrostatic potential distribution of 3C16 and 3C17.

9) Comparison of the present results with other similar findings in the literature should be discussed in more detail. This is necessary in order to place this work together with other work in the field and to give more credibility to the present results.

We have searched the relevant literature and compared the crystal structure with the relevant crystal structure in the literature. The supplementary content is located in section 3.1 of the revised manuscript and marked in red. The supplementary contents and references are as follows:

The two crystal structures have the same crystal system and space group as reported in the literature [41]. Unlike the compound reported in the literature, the number of mole-cules in a single crystal cell and the length of molecules are different.

10) Conclusion part is very long. Make it brief and improve by adding the results of your studies.

We have supplemented and optimized the conclusion. As follows:

 3C16 and 3C17 belong to the triclinic system with a space group P-1. It was discovered that H2O plays a vital role in securing the molecular frame-work of the two molecules. Hirshfeld surface analysis verified the presence of N-H...O intermolecular interaction with the amine donor and O-H...O intermolecular interac-tion with the H2O donor in both the two molecules. The 2D fingerprint indicated that the major contributions come from H...H (3C16 79.4%, and 3C17 80%) bonds. The void analysis showed that the mechanical properties of the two molecules are strong. The enrichment analysis indicated that these two kinds of intramolecular O-H contacts were powerful contact. A 3D energy framework construction revealed that dispersion energy was predominant in the two molecules. DFT calculations indicated that the experimental structural parameters are consistent with their theoretical counterparts. FMO analysis was used to determine the re-activity descriptors of the two molecules, and the charge distributions on the ESP dia-grams demonstrate the chemical reaction sites of the of the two molecules.

11) There are many grammatic mistakes. Improve the English grammar of the manuscript.

We have revised the grammatical errors in the full text and marked them in red in the revised manuscript.

We have resubmitted the manuscript to the editorial office of the International Journal of Molecular Sciences and we are very grateful to you for kind considering publication of the manuscript in your journal.

With my best regards,

Sincerely Yours,

Prof. Xin-Hui Fan

Round 2

Reviewer 1 Report

The authors have improved the manuscript in response to all the corrections and suggestions that were raised earlier. I am recommending the acceptance of the manuscript in its present form.